# Assessing self-selection biases in Facebook-recruited online surveys: Evidence from the COVID-19 Health Behavior Survey

Jessica Donzowa[1,2]*, Daniela Perrotta[1], Emilio Zagheni[1]

**1** Department of Digital and Computational Demography, Max Planck Institute for Demographic Research, Rostock, Mecklenburg Western Pomerania, Germany, **2** Faculty of Sociology, Bielefeld University, Bielefeld, Northrhine-Westphalia, Germany

* donzowa@demogr.mpg.de

**Data availability statement:** Due to privacy regulations and the potential risk of identifying respondents, the raw individual survey data

## Abstract

During the COVID-19 pandemic, many primary data collection efforts relied on online surveys via social media recruitment. According to the leverage-salience theory, respondents' varying interest in the survey topic can lead to differential survey responses, potentially introducing biases. In this study, we investigate the potential impact of displaying the survey topic in the survey recruitment materials on survey responses. We use data from the "COVID-19 Health Behavior Survey", a cross-national online survey that we ran between March and August 2020 in eight countries in Europe and North America (N=120,184). Respondents were recruited via targeted advertisements placed on Facebook with varying degrees of reference to the survey topic of COVID-19. The aim of our study is to assess whether stronger (or weaker) topic salience in the ad images is associated with higher (or lower) threat perceptions of COVID-19 and the adoption of preventive behaviors, including face mask use and increased hand-washing. Regression analyses show that in 20 of the 32 models, ad images had no significant effect on the survey outcomes. Factors like the month of survey participation or respondents' age were more influential. In the remaining models, where unexplained image effects persisted, the impact was minimal. While mask-wearing images were generally associated with lower threat perceptions of COVID-19 to oneself and the family, we found no consistent pattern for the adoption of protective behaviors. Overall, our findings do not provide consistent evidence that higher topic salience in our Facebook-based recruitment materials systematically influenced survey responses. However, in specific countries, certain recruitment images were linked to variations in COVID-19 threat perception and uptake of preventive behaviors. These context-specific effects highlight the importance of careful recruitment design for Facebook-based surveys during health crises.

used in the analysis cannot be shared publicly. The aggregated data underlying the main findings of this study, as well as the code used for the data cleaning, statistical modeling and the visualizations, which allows reproduction of the figures are available in the supplementary material. Interested researchers wishing to conduct scientific research can access the survey data upon request and approval from the COVID-19 Health Behavior Survey (CHBS). Contact details for accessing the data: datarequestCHBS@demogr.mpg.de.

**Funding:** This study was funded with support from the Max Planck Institute for Demographic Research, which is part of the Max Planck Society. The funders had no role in study design, data collection and analysis, decision to publish, or preparation of the manuscript.

**Competing interests:** The authors have declared that no competing interests exist.

## Introduction

Online surveys are widely used in social science because they offer a flexible and timely approach to data collection [1]. At present, a sizable fraction of these studies utilize Facebook by Meta Inc. as a source for recruiting respondents due to their broad coverage and the advantage to implement sampling quotas to target users with certain characteristics [2–4]. Especially during the COVID-19 pandemic, this approach was extensively used due to the impracticality of other recruitment methods (e.g., in-person interviews) to gain real-time information about crucial individual behaviors, such as compliance with health regulations or vaccination uptake, needed to inform health policies [5–8]. While many scholars have highlighted the advantages of using social media for survey recruitment, concerns have also been raised about data quality, often due to non-representativeness and self-selection biases based on interest in the survey topic [9–11]. In the case of survey recruitment via social media advertisements (e.g., Facebook advertisements), survey invitations are promoted through ad-hoc advertisements consisting of a short caption and an image or video. These elements, designed by the researcher, play a crucial role in conveying the survey topic during recruitment, potentially affecting self-selection issues [12]. While previous studies that used Facebook as a recruitment tool mainly focused on the effect of the advertisement design on participation rates and recruitment costs [10,13–16], only a few studies examined instead the impact of the survey topic salience in their advertisement designs on the survey outcomes. In a study on mental health among men in Australia, Choi et al. [17] found that the formulation "How tough is your mind?" received the highest number of clicks on Facebook, but these participants completed fewer questions about mental health. On the other hand, the participants recruited through the phrasing "Worried about your mental health?" reported poorer mental health status [17]. Stern et al. [18] used Facebook to recruit young men in the United States from sexual minorities, and found that image-based advertisements led to fewer non-substantive survey responses compared to video-based ads. In a study conducted in Turkey and Spain, Neundorf and Öztürk [19] found that advertisements with political references had lower recruitment costs, but also resulted in an unbalanced sample of participants with a heightened interest in politics. The most representative sample was obtained through advertisements mentioning an incentive [19]. In the context of the COVID-19 pandemic, while Facebook surveys provided timely and valuable insights, it remains unclear to what extent the survey data may have overestimated certain outcomes due to the self-selection of respondents with greater interest (or concern) in the pandemic.

In this paper, we seek to address this uncertainty by investigating whether varying levels of salience of the survey topic in Facebook advertisements during the recruitment process influence self-selection biases and, consequently, survey results. For this, we use data from the "COVID-19 Health Behavior Survey" (CHBS), a cross-national online survey that we conducted between March 13 and August 12, 2020, in eight countries in Europe and North America [20]. The questionnaire focused on respondents' behaviors and attitudes related to the COVID-19 pandemic. Recruitment took place via Facebook targeted advertisements that varied in the level of salience of the survey topic in the image shown in the advertisement, ranging from images that explicitly referred to COVID-19 to images that made no reference to the disease. The CHBS therefore offers quasi-experimental data that allow us to assess the impact of self-selection biases induced by topic salience on survey outcomes in the context of the COVID-19 pandemic.

We apply leverage-salience theory, as developed by Groves [21], to recruitment via Facebook advertisements. Specifically, we explore how displaying the survey topic in the most prominent part of the advertisements (i.e., the image) may influence substantive responses

to key survey outcomes relevant to inform health policy, such as the adoption of protective behaviors or the threat perception of the pandemic. According to the leverage-salience theory, Facebook users' decision to participate in a survey is influenced by the perceived benefits of participating (leverage), such as contributing to a survey topic of their interest or receiving a monetary incentive, as well as by how prominently these benefits are emphasized in the advertisement (salience) [21]. The aspects of the advertisement that seem beneficial to the Facebook user may vary depending on their individual characteristics [22]. This motivation driven by survey topic salience is also referred to as topical self-selection [23]. We argue that the level of salience of COVID-19 in our advertisement images may have differently influenced potential respondents' perceptions of the benefits of participation and, consequently, their motivation to participate. We assume that users who choose to participate in a survey on COVID-19 during the early months of the pandemic were specifically interested in the topic and, therefore, likely differed in their opinions and survey responses regarding the adoption of protective behaviors or threat perception in comparison to users that participated via an image referring to general health behavior. In addition to survey topic interest, several other factors may influence the choice to participate, such as age, gender, education, and the month of survey participation. The focus of this work is to explore how different elements, such as advertising content, participant demographics and timing of participation, shape survey responses related to the adoption of protective behaviors and threat perception. Our study expands on the previous research in several ways. First, while most of the existing studies focused on participation rates alone, here we focus on the effect of the advertisement content on the substantial survey responses. Moreover, while prior studies targeted a specific subpopulation (e.g., parents, smokers, or men), we targeted the general online population and controlled for additional demographic characteristics that may have affected the response behavior. Moreover, whereas previous studies focused on one country only, our study offers a cross-national comparison, and thus provides important insights for the design of future campaigns aimed at international recruitment.

## Materials and methods

### Study design

In this work, we use data from the "COVID-19 Health Behavior Survey" (CHBS), an online survey that we conducted via Facebook between March 13 and August 12, 2020, in seven European countries and the United States [20,24]. The survey data collection started on different dates across the various countries: March 13, 2020, in Italy, the United Kingdom, and the United States; March 17, 2020, in Germany and France; March 19, 2020, in Spain; April 1, 2020, in the Netherlands; and April 2, 2020, in Belgium. The data collection ended in all countries on August 12, 2020. The questionnaire was designed to explore people's behaviors and attitudes related to the COVID-19 pandemic (e.g., threat perceptions, uptake of preventive measures), along with socio-demographic characteristics (e.g., age, sex, and education) and health indicators (e.g., underlying medical conditions). The questionnaire was available in both English and the national language(s) of the respective countries. The full questionnaire in English can be found in [20]. Ethics approval was received from the Ethics Council of the Max Planck Society (Application No: 2020_07) on April 2, 2020. Electronic informed consent was obtained from all participants who actively opted in to participate in the online survey by clicking a "Continue" button that confirmed they were willing to participate in the survey, were at least 18 years old, and had read the data protection policy.

Respondents were recruited via targeted advertisements on Facebook. The central part of the advertisement consists of a visual element, either a video or as in our case an image, which

can vary in their association to the survey topic from no reference to high topic salience. Additionally the advertisement consists of a short caption and a link to the questionnaire, which is hosted on an external web page. The advertisements are hosted by a Facebook page, which is used to provide additional background information about the research, and thus to build trust with potential respondents [12]. We implemented targeting criteria with the aim of disseminating our survey homogeneously across different demographic groups, thus limiting the bias that Facebook's advertising algorithms may generate in the recruiting process when optimizing the advertisement campaigns. For this, we stratified our advertisements by sex (i.e., male and female), age group (i.e., 18–24, 25–44, 45–64, and 65+ years), and region of residence (as inferred by Facebook). In the European countries, the region classification largely followed the NUTS-1 classification, which we aggregated into larger macro-regions. In the United States, the region classification was based on census regions. More details on region stratification used in the Facebook advertising campaigns can be found in [20].

Each advertisement displayed one image selected by Facebook's algorithms from a total of six different advertisement images (Fig 1). Given the unique circumstances of the COVID-19 pandemic and the urgency of launching the survey as quickly as possible, the images used in the survey recruitment phase of our study are not validated measures of salience. However, the images can be grouped into three categories based on how prominently they referred to the survey topic of the COVID-19 pandemic. Images 1 and 2 are fairly neutral, showing individuals exercising outdoors. Images 3 and 4 referred to respiratory illnesses but not to COVID-19 specifically. Lastly, images 5 and 6 explicitly hinted at the survey topic by

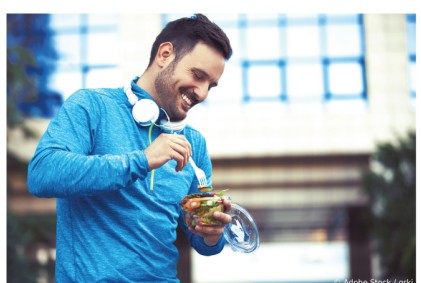

(1) Male athlete
©Adobe Stock/grki

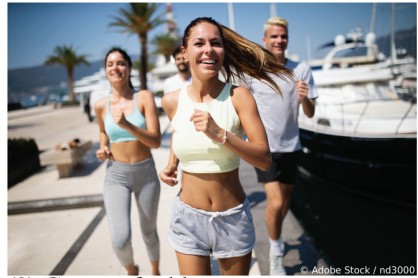

(2) Group of athletes
©Adobe Stock/nd3000

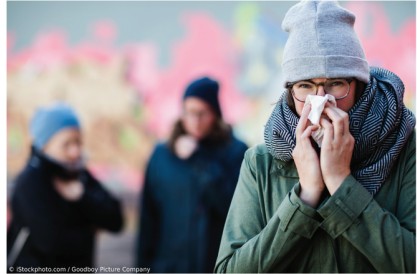

(3) Woman blowing nose
©iStockphoto/Goodboy Picture Company

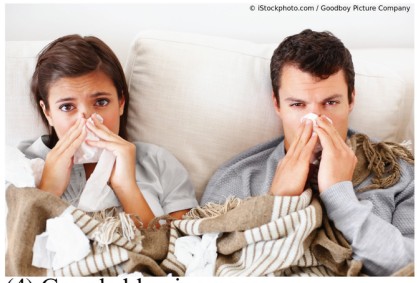

(4) Couple blowing nose
©iStockphoto/Goodboy Picture Company

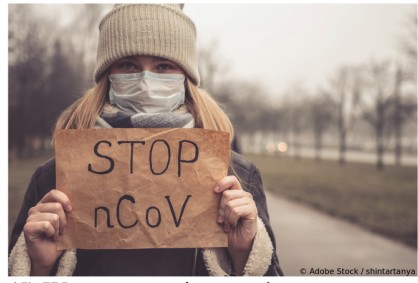

(5) Woman wearing mask
©Adobe Stock/shintartanya

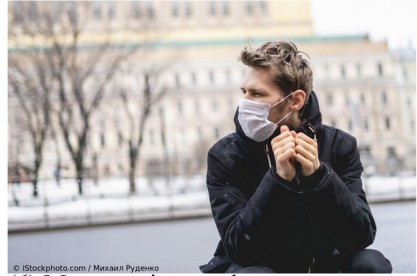

(6) Man wearing mask
©iStockphoto/Mihail Rudenko

**Fig 1. Images used in the Facebook advertisements to recruit respondents for the "COVID-19 Health Behavior Survey" conducted between March and August 2020 in eight countries in Europe and North America.**

portraying individuals wearing a protective face mask. This is because in the countries of this study, the use of protective face masks was a public health recommendation specifically in response to COVID-19. This measure was still relatively new and contextually specific to COVID-19, particularly in Western countries. Image 5, in particular, explicitly referred to COVID-19 by displaying an early name of the virus. It is important to note that the only reference to COVID-19 was made through the advertisement images, whereas the advertisement caption and Facebook page referred to "health behavior" but not specifically to COVID-19. Based on the leverage-salience theory, we expect that, depending on the respondents' interest in the survey topic of COVID-19, the various images had a different potential to generate topical self-selection bias, ranging from low potential (images 1 and 2) to medium (images 3 and 4) to high (images 5 and 6). We are able to investigate this because, for each survey participant, we collected the information about the specific advertisement (and therefore about the advertisement image) that directed them from Facebook to our survey. It is important to note that in our study the selection of the advertisement image was controlled by Facebook's algorithms in order to optimize advertisement delivery and enhance the likelihood of user engagement according to the campaign objective. We used the campaign objective "traffic", which means the algorithm had the goal of promoting advertisements to users more likely to click on the link in the advertisement. Previous research has shown that "reach" campaigns are not suitable for survey recruitment, while "conversion" campaigns are most efficient. "Traffic" and "conversion" campaigns do not differ significantly from each other in terms of demographic representativeness [25]. However, a "conversion" campaign requires the implementation of the so-called Meta pixel, which provides Meta with additional information about the respondent's behavior outside of the Facebook platform, in order to optimize for survey completion [26]. For research ethics reasons, we choose to avoid this additional tracking and choose the "traffic" objective. We report the number of link clicks that each advertisement image received and the number of Facebook users who saw the advertisement image (called "reach") to provide an overview of the user engagement resulting from the advertisement placement. Both metrics are part of the campaign performance data provided by Meta. For the interested reader, more details on the design of the Facebook advertising campaigns can be found in the following references: [3,20,24].

## Statistical modeling

In this study, we employ generalized linear models to assess the potential impact of the advertisement images on the survey responses. In particular, here we examine the adoption of preventive measures and levels of threat perception, while also accounting for socio-demographic differences. By analyzing the levels of threat perceived by respondents, we can contrast them with the different levels of salience that the advertisement images may have generated during recruiting phase and, consequently, with the levels of uptake of protective behaviors towards COVID-19. In this way, our study offers an insightful case study on the reliability of online surveys for informing policies and interventions during health crises.

Empirical evidence suggests that the level of concern regarding a COVID-19 infection differed across demographic subgroups, with sex, age, and education being among the most important determinants. The highest levels of concern towards COVID-19 were found among women [27,28] and the elderly [29] compared to men and younger individuals. As for educational attainment, individuals with a lower level of education were less likely to comply with COVID-19 containment measures, such as avoiding large gatherings or increasing hand hygiene, compared to individuals with a higher level of education [30]. Our previous studies

using the CHBS data confirm these findings showing higher perceived threat of COVID-19 among women and respondents over the age of over 65 years [20,31].

In light of this evidence, we include the age, sex, and educational attainment of respondents as control variables in our models in order to account for the effects that the advertisement image may have on the sample composition. Moreover, to account for changes in the COVID-19 pandemic as well as in public health interventions over time, we include the month of survey participation as control variable. Specifically, the independent variables in our models correspond to the following categories:

- *Image* refers to the advertisement image through which a Facebook user reached our survey. We used image 3 ("Woman blowing nose") as the reference.
- *Sex* refers to the respondents' self-reported sex (i.e., male, female). We used female as the reference.
- *Age* represents the respondents' age, grouped into four age brackets, i.e., 18-24, 25-44, 45-64, and 65+. We used the 18-24 age group as the reference.
- *Education* refers to the respondents' educational attainment, grouped into three categories: "secondary school or lower", "university level", and "postgraduate degree". We used "secondary school or lower" as the reference. The detailed breakdown and the re-coding of the education categories can be found in S1 Table in S1 File).
- *Month* refers to when the respondent participated in our survey, ranging from March to August 2020. We used March as the reference in all countries, except for Belgium and the Netherlands, where we used April as the reference due to a later start of the survey data collection.

To evaluate the influence of the advertisement images on the respondents' threat perception, we employed a multinomial logistic regression analysis. Here we considered the respondents' perception of the threat that COVID-19 posed to both themselves and their family members. In the questionnaire, respondents were asked to rate the levels of perceived threat on a five-point Likert-scale, ranging from "very low threat" to "very high threat". We consolidated this scale into a variable with three possible values (low, medium, and high) by combining the categories "very low threat" and "low threat", and "very high threat" and "high threat". By using the "high" category as a base category, the model takes the following forms:

$$\ln \frac{P_{\text{Low}}}{P_{\text{High}}} \quad = \quad \beta_0 + \beta_{\text{image}} + \beta_{\text{sex}} + \beta_{\text{age}} + \beta_{\text{education}} + \beta_{\text{month}} + e_i \tag{1}$$

$$\ln \frac{P_{\text{Medium}}}{P_{\text{High}}} \quad = \quad \beta_0 + \beta_{\text{image}} + \beta_{\text{sex}} + \beta_{\text{age}} + \beta_{\text{education}} + \beta_{\text{month}} + e_i \tag{2}$$

To predict the adoption of preventive behaviors, instead, we used a binomial logistic regression model. We examined the use of protective face masks and the increased frequency of hand washing (compared to the pre-pandemic period). This was measured in the survey as a dichotomous variable with categories "yes" and "no". Due to some changes in the CHBS questionnaire implemented on May 7, 2020, both survey questions were slightly reworded, but with no substantive change in meaning (see S2 Table in S1 File). It is important to note that while wearing a face mask was strongly influenced by public health interventions, hand hygiene was more related to individuals' changes in behaviors. The model takes the following form:

$$\log \frac{P}{1-P} \quad = \quad \beta_0 + \beta_{\text{image}} + \beta_{\text{sex}} + \beta_{\text{age}} + \beta_{\text{education}} + \beta_{\text{month}} + e_i \tag{3}$$

where *P* refers to the probability of having adopted the given behavior, i.e., wearing a protective face mask or increasing the frequency of hand hygiene.

For a visual summary of our study design, please refer to S1 Fig in the S1 File. In both models, the regression analysis calculates separate step-wise models for each country and survey outcome. Considering the four outcomes across eight countries, this results in 32 final models, each including all covariates. The alpha level of 0.05 is used to determine statistical significance. All final models were tested for multicollinearity using the variance inflation factor indicating no multicollinearity among the predictor variables. Out of a total of 289,973 submitted questionnaires, we included in our analyses only 120,184 questionnaires that i) were complete, ii) had no missing information on the advertisement image through which respondents entered our survey. This is only possible for respondents that reached our survey via the targeted advertisements. However, respondents could still access our survey through other means, for example, if the advertisement appeared in their feed after being shared by other users, or if someone shared the link to the survey. Finally iii) those which had informative responses in both dependent and independent variables (i.e., excluding non-informative responses, such as "Don't know" and "Prefer not to answer") (see S2 Fig in S1 File for an overview of the number of questionnaires excluded in each data preparation step). Data analysis and modeling was conducted using R version 4.3.2. For the code used for the data cleaning, statistical modeling, and the visualizations, see S3 File. For the aggregated data underlying the main findings of this study, see S4 File.

## Results

In the following, we will first provide descriptive statistics on the sample composition and survey completion, as well as survey outcomes by advertisement image and the socio-demographic characteristics of the sample. Next we present the model results on the probability of threat perceptions and the adoption of preventive measures based on the advertising image through which respondents entered the survey.

### Sample composition by advertisement image

Our dataset consists of 120,184 completed questionnaires from Belgium (N = 11,337), France (N = 11,791), Germany (N = 22,518), Italy (N = 14,397), the Netherlands (N = 8,364), Spain (N = 11,280), the United Kingdom (N = 11,500), and the United States (N = 28,997). First we look at number of link clicks that the different advertisement images received on Facebook.

The highest number of link clicks was received from the "woman wearing mask" image (image 5) in all countries, ranging from 63.9% in the United States to 28.1% in Spain. The "male athlete" image (image1) (Spain: 7.6% and France: 0.4%) and the "couple blowing nose" image (image 4) (Spain: 4.9% and Germany/France: 1.9%) received the lowest number of link clicks. The advertisement reach, which refers to the number of unique Facebook users who saw the advertisement, follows the same pattern by image (S3 Table in S1 File).

Next we calculated the survey completion rate as the number of completed questionnaires divided by the number of link clicks that each advertisement received. This measure indicates how efficiently the number of received link clicks by a Facebook user convert into a survey respondent. The mask-wearing images (images 5 and 6) had the highest survey completion rates in all countries, ranging from 2.7% for the "man wearing mask" image (image 6) in Spain to 17% in the United States for the "woman wearing mask" image (image 5). The completion rate was the lowest for the athlete images (images 1 and 2). It ranged from 0.2% in Spain to 5% in the United States. For the nose-blowing images (images 3 and 4), the

completion rate ranged from 1.1% in Spain for the "woman blowing nose" image (image 3) to 8.2% in France for the "couple blowing nose" image (image 4) (see Table 1).

As a consequence of the completion rate, the highest share of participants in our final sample reached our survey through the "woman wearing mask" image (image 5), ranging from 56% in the Netherlands to 81% in the US. This was followed by the "man wearing a mask" image (image 6), from 10% of participants in Italy to 27% of participants in Belgium. For the "woman blowing nose" image (image 3) the share ranged between 18% in the Netherlands to 3% in the United States. The "couple blowing nose" image (image 4) made up between 3% of the participants in Spain and 1% in Germany. Finally, the images displaying athletes (images 1 and 2) recruited only a very small share of participants, ranging from 0.1% in France to 2% in Spain (see Table 1).

Looking at this over time, the mask-wearing images (images 5 and 6) and the "woman blowing nose" image (image 3) recruited the majority of participants during the first months of the campaign in March and April 2020. The remaining images recruited a stable number of participants over time. Notably, the number of participants recruited through the athlete images (images 1 and 2) never surpassed 100 participants per month (see Fig 2). Because of these low numbers, we excluded the two athlete images from the multivariate analysis.

Regarding the overall sample composition, the share of women was higher than for men in all countries, ranging from 60% in Germany to 70% in France. Respondents' median age

**Table 1. Survey completion rate and distribution of completed surveys by country and advertising image.**

| Country | IMG 1 Male athlete | IMG 2 Group of athletes | IMG 3 Woman blowing nose | IMG 4 Couple blowing noses | IMG 5 Woman wearing mask | IMG 6 Man wearing mask | Country total |
|---|---|---|---|---|---|---|---|
| **Completion rate in %** | | | | | | | |
| BE | 3.0% | 2.9% | 4.6% | 5.1% | 16.8% | 10.9% | 11.4% |
| FR | 1.3% | 1.1% | 2.4% | 8.2% | 13.2% | 8.9% | 9.8% |
| DE | 3.3% | 1.5% | 1.4% | 5.5% | 13.3% | 7.8% | 9.1% |
| IT | 0.8% | 0.4% | 2.2% | 3.8% | 7.4% | 5.1% | 5.0% |
| NL | 1.1% | 1.7% | 2.5% | 3.1% | 7.8% | 6.3% | 5.1% |
| ES | 0.3% | 0.2% | 1.1% | 1.2% | 4.9% | 2.7% | 2.2% |
| UK | 1.1% | 1.1% | 4.1% | 4.9% | 9.2% | 8.5% | 7.8% |
| US | 5.2% | 1.2% | 7.6% | 7.1% | 17.1% | 14.2% | 13.4% |
| **Number of completed surveys (%)** | | | | | | | |
| BE | 38 | 155 | 877 | 181 | 6,981 | 3,105 | 11,337 |
| | (0.3%) | (1.4%) | (7.7%) | (1.6%) | (61.6%) | (27.4%) | (100.0%) |
| FR | 6 | 115 | 427 | 189 | 9,569 | 1,485 | 11,791 |
| | (0.1%) | (1.0%) | (3.6%) | (1.6%) | (81.2%) | (12.6%) | (100.0%) |
| DE | 45 | 110 | 776 | 136 | 17,964 | 3,360 | 22,518 |
| | (0.2%) | (0.5%) | (3.4%) | (1.2%) | (79.8%) | (14.9%) | (100.0%) |
| IT | 47 | 148 | 1,154 | 309 | 11,238 | 1,501 | 14,397 |
| | (0.3%) | (1.0%) | (8.0%) | (2.1%) | (78.1%) | (10.4%) | (100.0%) |
| NL | 20 | 121 | 1,519 | 220 | 4,663 | 1,821 | 8,364 |
| | (0.2%) | (1.4%) | (18.2%) | (2.6%) | (55.8%) | (21.8%) | (100.0%) |
| ES | 137 | 179 | 1,133 | 299 | 7,052 | 2,480 | 11,280 |
| | (1.2%) | (1.6%) | (10.0%) | (2.7%) | (62.5%) | (22.0%) | (100.0%) |
| UK | 24 | 75 | 876 | 259 | 8,630 | 1,639 | 11,500 |
| | (0.2%) | (0.7%) | (7.6%) | (2.3%) | (75.0%) | (14.2%) | (100.0%) |
| US | 199 | 398 | 840 | 395 | 23,594 | 3,571 | 28,997 |
| | (0.7%) | (1.4%) | (2.9%) | (1.4%) | (81.4%) | (12.3%) | (100.0%) |

Table notes The survey completion is calculated as the number of completed questionnaires divided by the number of unique Facebook users who clicked on the advertisement. For brevity, we used country codes: BE (Belgium), FR (France), DE (Germany), IT (Italy), NL (Netherlands), ES (Spain), UK (United Kingdom), and US (United States).

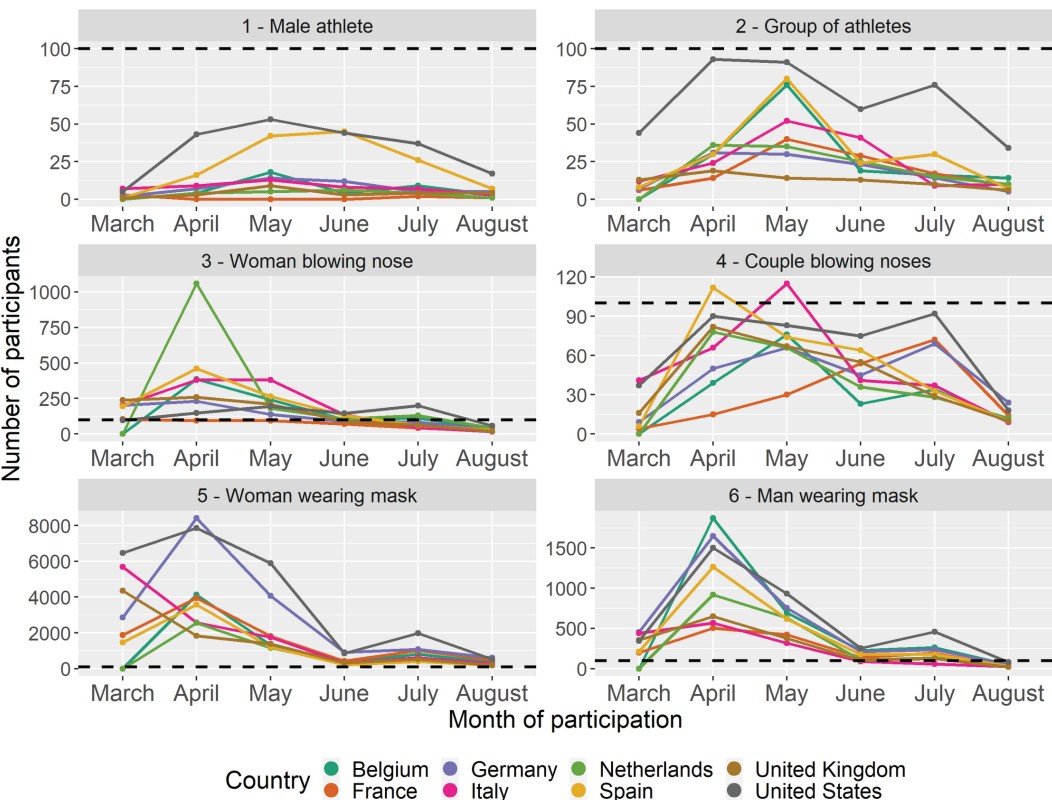

**Fig 2. Number of participants by advertisement image and country over the survey period.** Please note the different scales on the y-axis due to the large differences in participation by image. The horizontal line marks 100 participants.

varied, with the lowest in Germany and Italy at 42 years and highest in the United States and the United Kingdom at 58 years. Respondents with the largest share of university-level education came from France (70%), Spain (56%), and the United States (56%). In Germany (64%), Italy (55%), and the Netherlands (74%) the largest percentage of respondents had a secondary school or lower level education. In Belgium and the United Kingdom, both secondary school or lower education and university education account for approximately 46% to 48% in both countries (Table 2). A more detailed description of the covariates by ad image, as well as impressions by age, gender and ad image, can be found in the supporting information S1 File (see S4 to S7 Tables in S1 File). In general, the age and gender composition by ad image is consistent with the number of impressions (i.e., the number of times the ad was displayed) recorded for the ad sets targeted by gender and age.

## Survey outcomes by advertisement image

Fig 3 shows the variation in the levels of perceived threat towards COVID-19 that respondents reported for themselves and for their family, across the various countries and advertisement images. Respondents generally perceived COVID-19 as a higher threat to their family than to themselves. The athlete images (images 1 and 2) and the "couple blowing nose" image (image 4) tended to recruit respondents with a lower threat perceptions. Except in France, where the "male athlete" image (image 1) resulted in the highest threat perception, although this only refers to a sample size of 6 respondents. We do not observe striking differences in the

**Table 2. Sample composition by country, sex, age, and education.**

|  | BE | FR | DE | IT | NL | ES | UK | US |
|---|---|---|---|---|---|---|---|---|
| Female respondents (%) | 65.5 | 69.5 | 59.7 | 66.5 | 62.3 | 68.0 | 64.5 | 63.6 |
| Respondents' median age (IQR) | 51 | 48 | 42 | 42 | 57 | 52 | 58 | 58 |
|  | (34-63) | (30-62) | (30-58) | (29-58) | (41-65) | (39-61) | (43-66) | (40-67) |
| **Educational status (%)** |  |  |  |  |  |  |  |  |
| Secondary or lower | 48.4 | 22.7 | 64.2 | 55.0 | 73.9 | 33.8 | 45.5 | 38.4 |
| University | 48.7 | 70.8 | 32.5 | 35.3 | 24.4 | 56.4 | 46.8 | 56.1 |
| Postgraduate | 3.0 | 6.5 | 3.3 | 9.8 | 1.7 | 9.8 | 7.8 | 5.5 |

Table notes For brevity, we used country codes: BE (Belgium), FR (France), DE (Germany), IT (Italy), NL (Netherlands), ES (Spain), UK (United Kingdom), and US (United States).

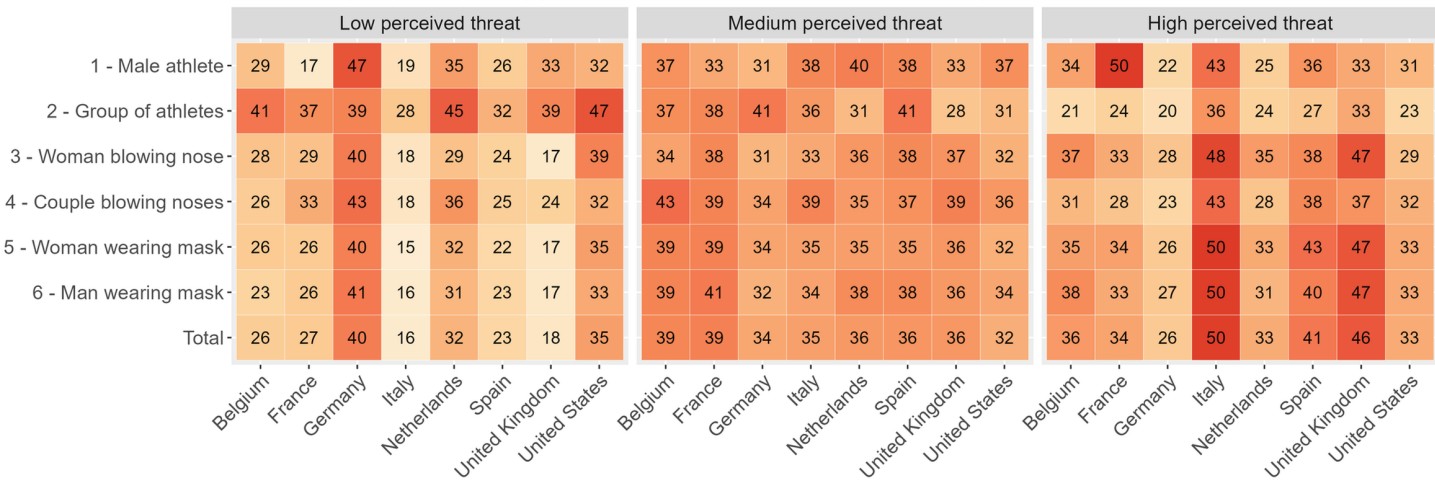

**Fig 3. Respondents' threat perception of COVID-19 to the family and to themselves, by advertisement image and country.**

threat perceptions among the respondents recruited through the mask-wearing images (image 5 and 6).

Fig 4 shows the adoption rates of preventive measures, i.e., wearing a face mask and increased hand washing, across the various countries and advertisement images. The use of face masks was lowest in the Netherlands (14%) and the United Kingdom (27%), and highest in Spain (70%) and Italy (73%). In all countries, more than 87% of participants increased the frequency of hand washing as a protective measure to the pandemic. In most countries, the respondents recruited through the athlete images (images 1 and 2) and the nose-blowing images (images 3 and 4) reported higher adoption of face masks, except in France for the "male athlete" image (image 1) and in the Netherlands for the "woman blowing nose" image (image 3). However, for France it is again important to note that the "male athlete" image (image 1) only refers to a sample size of 6 respondents. The level of compliance with hand

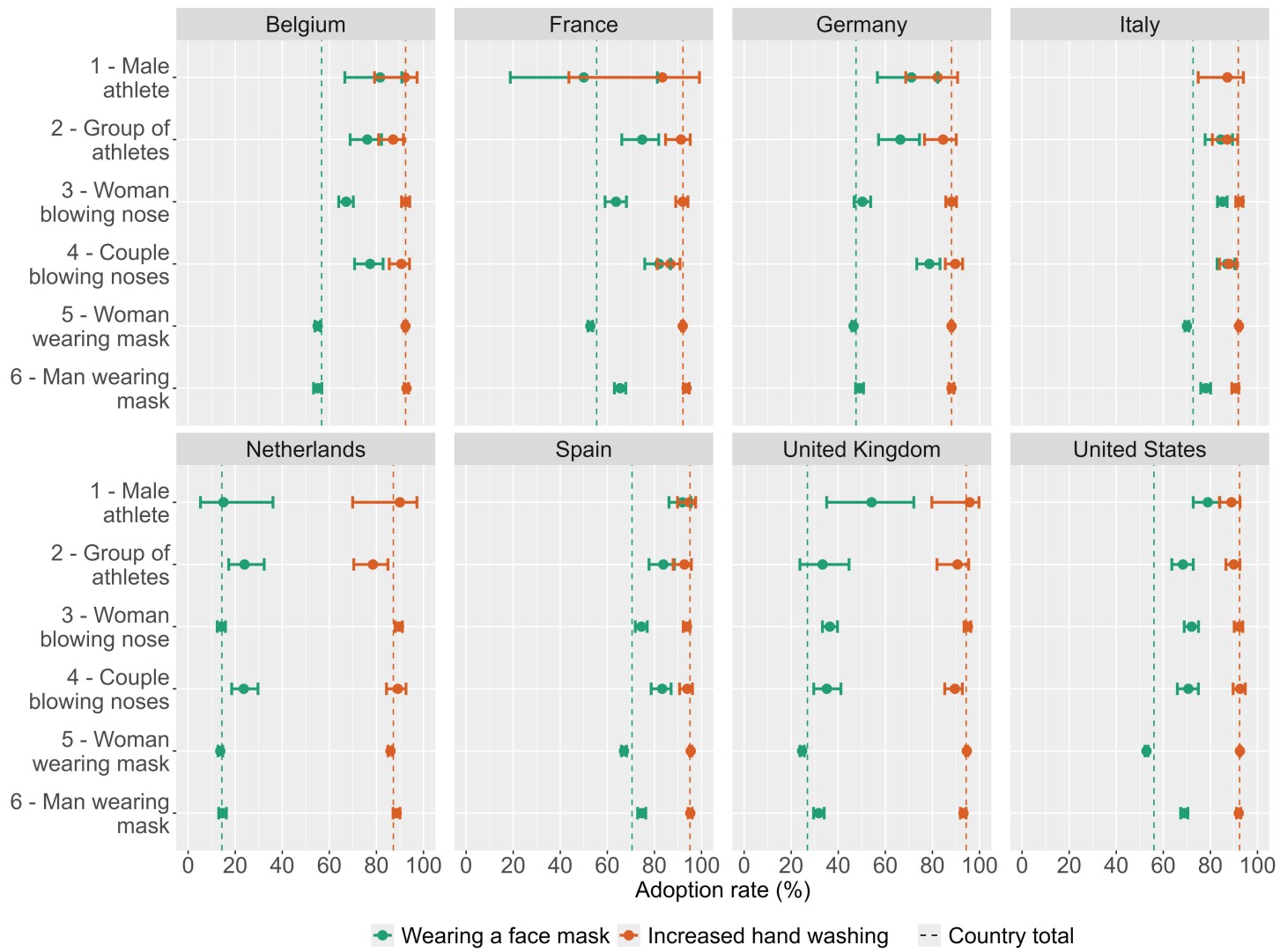

**Fig 4. Adoption rates of preventive measures by advertisement image and country.** The plots show the proportion of respondents who adopted the behavior and the 95% Wald confidence interval. The vertical lines indicate the adoption rates at the country level.

hygiene recommendations only differs in the Netherlands for the "group of athletes" image (image 2), for which respondents demonstrated a lower likelihood of increasing the frequency of hand washing.

### Impact of advertisement images on threat perceptions of COVID-19

Fig 5 shows the resulting model estimates of the predicted probability of threat perception of COVID-19 to the family and to oneself, using multinomial logistic regression models, broken down by advertisement image and country. Note that for this part of the analysis, we excluded images 1 and 2 (i.e., "male athlete" and "group of athletes") because of the low numbers of

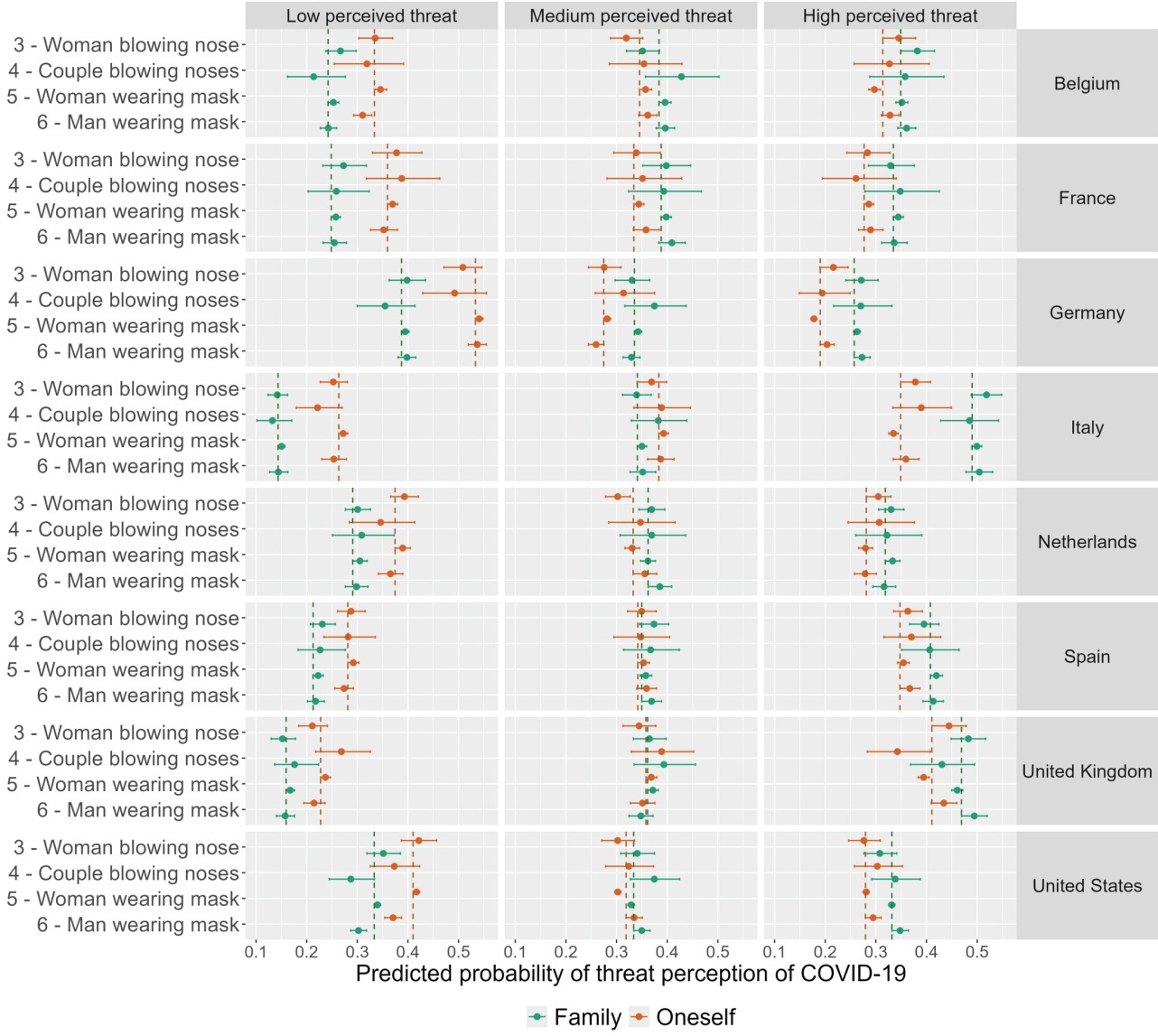

**Fig 5. Predicted probability of the perceived threat of COVID-19 to the family (in green) and to oneself (in orange) by advertisement image (on the y-axis) and country.** Controlled for sex, age, education, and month of survey participation. The vertical line corresponds to the lowest confidence interval (CI) and serves as a visual cue to identify overlapping confidence intervals.

respondents who reached our questionnaire through the corresponding advertisements on Facebook (see Section Sample composition by advertisement image for more details). Overall the model estimates confirm the pattern that we observed looking solely at descriptive statistics on how the threat perception is generally higher to family members than to oneself.

The threat perception of COVID-19 to the family, only differed across advertisement images in the United States. Here, the respondents recruited through the "woman wearing mask" image (image 5) were more likely to perceive a low threat (34%, CI: 0.33-0.35) compared to those recruited through the "couple blowing nose" (29%, CI: 0.24-0.33) image (image 4) and the "man wearing mask" image (image 6) (30%, CI: 0.29-0.32). The perceived threat of COVID-19 to the family did not differ across advertisement images in the remaining countries. Specifically, the images had no effect on the perceived threat of COVID-19 to the family in any of the models in Belgium, France, Germany, Italy, and the United Kingdom. Initial differences were explained by adding the age to the model in Spain, and the month of survey participation in the Netherlands (see S4 and S5 Figs in S1 File).

On the contrary, the perceived threat of COVID-19 to oneself varied based on the advertising images in Belgium, Germany, Italy, the United Kingdom, and the United States. In both Germany and the United States, the image "woman wearing mask" (image 5) (Germany: 18% (CI: 0.17-0.18); United States: 37% (CI: 0.35-0.39)) was associated with a lower threat perception compared to the image "man wearing mask" (image 6) (Germany: 20% (CI: 0.19-0.22); United States: 42% (CI: 0.41-0.42). In Italy, participants shown the "woman blowing nose" image (image 3) (38%, CI: 0.35-0.41) were more likely to perceive a high threat compared to those recruited through the "woman wearing mask" image (image 5) (33%, CI: 0.33-0.34). In the United Kingdom, both the "woman blowing nose" image (image 3) (44%, CI: 0.41-0.48) and the "man wearing mask" image (image 6) (43%, CI: 0.41-0.46) were more likely to perceive a high threat compared to the "couple blowing nose" image (image 4) (34%, CI: 0.28-0.41) and the "woman wearing mask" image (image 5) (39%, CI: 0.38-0.41). Finally, we observed two effects in Belgium: i) the respondents recruited through the "woman wearing mask" image (image 5) were associated with a lower threat perception of COVID-19 (35%, CI: 0.33-0.36) compared to those recruited through the "man wearing mask" image (image 6) (31%, CI: 0.29-0.33), and ii) the respondents recruited through the "woman blowing nose" image (image 3) (35%, CI: 0.31-0.38) were more likely to perceive COVID-19 as a high threat compared to those from the"woman wearing mask" image (image 5) (30%, CI: 0.29-0.31).

We found no effect of the advertising image on the respondents' perceived threat of COVID-19 to themselves in any of the models in Spain. Initial differences were explained by adding the month of survey participation in the model in the Netherlands and France (see S6 and S7 Figs in S1 File).

## Impact of advertisement images on respondents' adoption of preventive behaviors

Fig 6 shows the resulting model estimates of the predicted probability, estimated from the binomial logistic regression model, of wearing a face mask and of increased hand washing by advertisement image and country. We found variations in the probability of wearing a face mask based on the advertisement images in Belgium, Germany, the United Kingdom, and the United States, while significant differences in increased hand hygiene were observed only in Italy and the United Kingdom, but with smaller effect sizes.

As for the probability of wearing a face mask, in Belgium, the respondents recruited through the "couple blowing nose" image (image 4) (70%, CI: 0.61-0.77) were more likely to wear a face mask compared to those recruited through the "man wearing mask" image

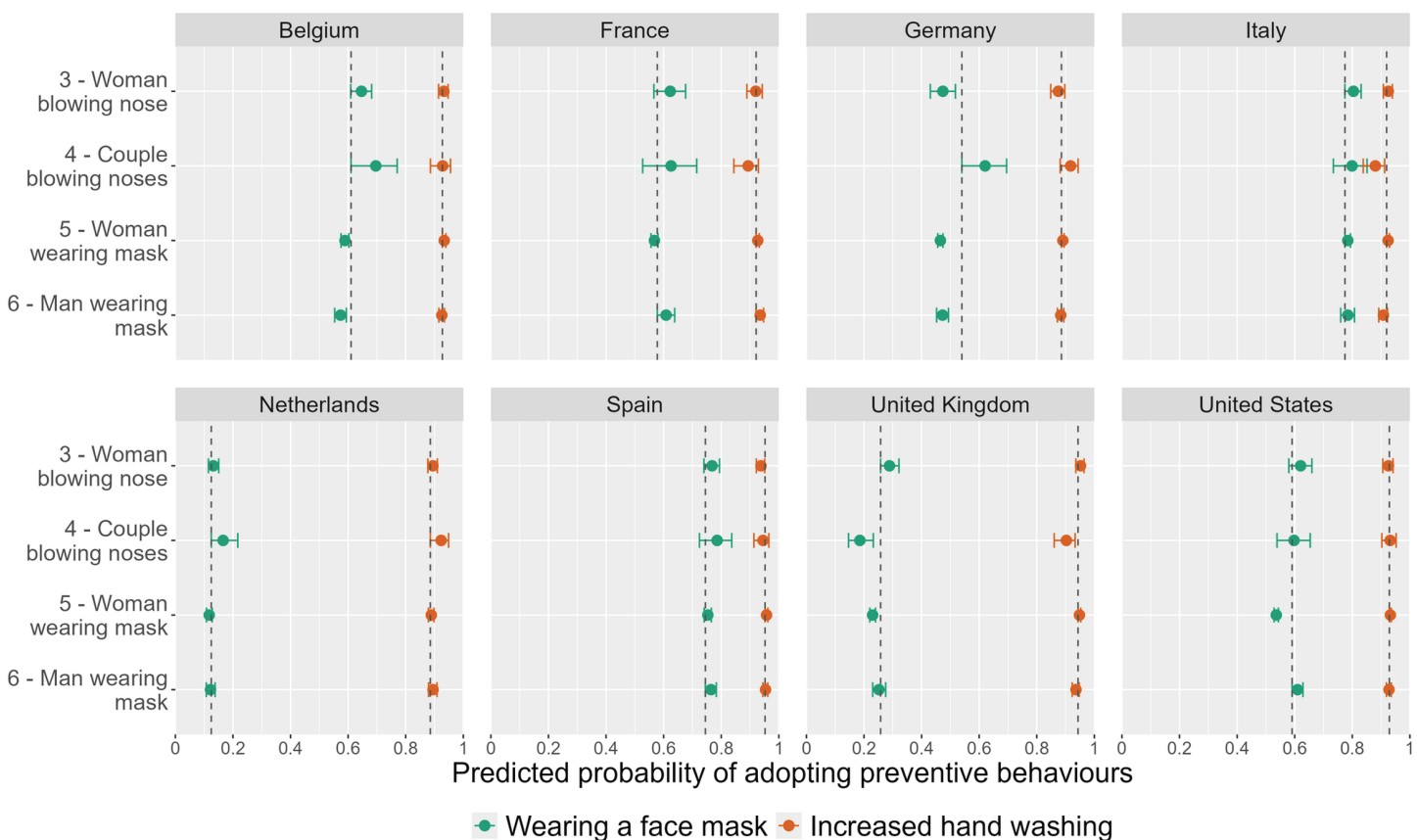

**Fig 6. Predicted probability of wearing a face mask (in green) and increased hand washing (in orange) by advertisement image (on the y-axis) and country.** Controlled for sex, age, education, and month of survey participation. The vertical line corresponds to the lowest confidence interval (CI) and serves as a visual cue to identify overlapping confidence intervals.

(image 6) (57%, CI: 0.55-0.59). In Germany, the respondents recruited through the "couple blowing nose" image , had a higher probability of wearing a face mask as well with 62% (CI: 0.54-0.70) compared to all other images with 47%. In the United States, a higher chance of wearing a face mask was associated with the images "man wearing mask" image (image 6) (61%, CI: 0.60-0.63) and "woman blowing nose" image (image 3) (62%, CI: 0.58-0.66) compared to the "woman wearing mask" image (image 5) (54%, CI: 0.53-0.54). In contrast to the other countries in the United Kingdom, a higher chance of wearing a face mask was associated with the "woman blowing nose" image (29%, CI: 0.26-0.32) compared to the "couple blowing nose" image (image 4) (19%, CI: 0.15-0.23) and the "woman wearing mask" (image 5) (23%, CI: 0.22-0.24). Lastly, the differences across images in France, Italy, Spain and the Netherlands were explained by adding the month of survey participation to the model (see S8 and S9 Figs in S1 File).

Looking at the probability of increased hand hygiene in Fig 6, we found significant differences across images only in Italy, and the United Kingdom, with smaller effect sizes compared to the use of face masks. Contrary to its effect on the face mask wearing, the "couple blowing nose" image (image 4) was associated with less hand hygiene compared to the mask-wearing images (images 5 and 6). In Italy, the participants recruited through the "couple blowing nose" image (image 4) were less likely (88%, CI: 0.84-0.91) to increase the hand washing

frequency in comparison to those recruited through the "woman wearing mask" image (image 5) (92%, CI: 0.92-0.93). Similarly, in the United Kingdom, respondents recruited through the "couple blowing nose" image (image 4) had a 90% chance (CI: 0.86-0.93) of increased hand hygiene frequency compared to 95% (CI: 0.94-0.95) for those recruited through the "woman wearing mask" image (image 5). Finally, we found no significant effect of the advertisement images on the survey measure of increased hand hygiene in Belgium, Germany, Spain, and the United States. Initial differences were explained by adding the age to the model in France and the sex in the Netherlands (see S10 and S11 Figs in S1 File). Complete step-wise regression tables for all models, displaying the Odds Ratio, standard error, confidence interval and p-value can be found in S2 File.

We also tested the inclusion of respondents' employment status over the past seven days in our model and found that our results remain robust, showing no differences compared to the model without employment status. While exploring the interaction between the month of survey participation and the ad image would be interesting for our research question, small case numbers or empty cells in the cross-tabulation between our outcome variables (image and month of survey participation) prevent us from estimating this interaction with sufficient precision.

To summarize, in 20 out of our 32 models we found no image effect in the final model after adjusting for sex, age, education and month of survey participation. Initial differences between the images in threat perception or preventive behaviors were most often explained by the respondents' age or the month of survey participation. Overall, when unexplained differences between images remained, the effect sizes were small and no systematic direction could be detected.

## Discussion

The goal of this work was to investigate whether the survey responses differed depending on the image used in the advertisement to recruit respondents via Facebook. According to the leverage-salience theory, differential levels of interest in the survey topic are associated with differential motivation to survey participation and subsequent answering behaviors. We leveraged unique survey data from the "COVID-19 Health Behavior Survey", a cross-national online survey that we ran via Facebook between March and August 2020 in eight countries in Europe and North America. The survey respondents were recruited via targeted advertisements on Facebook that differed only in the advertisement image, chosen by Facebook algorithms among six different images with different levels of reference to the survey topic. We used regression analyses to examine the effects of the advertisement images on the probability of the threat perceptions of COVID-19 and the adoption of preventive measures. We hypothesized that images with stronger reference to the COVID-19 pandemic would lead to a selection of respondents with greater concern in the survey topic, and thus with higher threat perceptions of COVID-19 and higher levels of compliance with recommended health behaviors compared to respondents recruited through images that made little to no reference to the survey topic.

The descriptive analysis showed that survey completion rates were highest for the mask-wearing images (images 5 and 6) and lowest for the athlete images (images 1 and 2). Secondly, the respondents recruited through the athlete images (images 1 and 2) and the "couple blowing nose" image (image 4) tended to perceive the threat of COVID-19 as lower. This confirms previous descriptive research using the CHBS survey data that showed variations in the perceived threat level of COVID-19 by advertisement image [24]. Regarding preventive measures, we observed that respondents recruited through the mask-wearing images (images

5 and 6) were less likely to use a face mask. The level of compliance with hand hygiene recommendations only differed in the Netherlands for the "group of athletes" image (image 2), with a lower likelihood of hand washing.

Our findings from the regression analysis revealed that the impact of the advertisement images on the various survey outcomes was explained by adding either the month of survey participation or the respondents' age to the model. Specifically, the month of survey participation was a significant explanatory factor for i) the use of a protective face mask in France, Italy, the Netherlands, and Spain, ii) the threat perception of COVID-19 to the family in the Netherlands and France and iii) the threat perception of COVID-19 to oneself in the Netherlands. Respondents' age explained the image effect on the threat perception of COVID-19 the family in Spain, as well as increased hand washing in France.

Secondly, in those cases where unexplained image effects on the survey outcomes persisted, the direction of these effects differed depending on the specific outcome and country. Contrary to our initial hypothesis, we generally found that higher salience of the survey topic in the advertisement image was associated with lower compliance to wear a face mask. However, this association also displayed variations across countries, for example, the "couple blowing nose" image (image 4) was associated with a higher probability of wearing a face mask in Germany and a lower probability in the United Kingdom. Conversely, respondents recruited through mask-wearing images were more likely to report increased hand washing. Regarding the threat perception of COVID-19 to oneself and to the family (only for the United States), respondents recruited through the "woman wearing mask" image (image 5) tended to have a lower threat perception, although the effect sizes were small.

To provide some context, we interpret our results in light of the different public health interventions and the progression of the COVID-19 pandemic which varied across the countries and over time [32,33]. Considering the first phase of the pandemic in spring and summer 2020, the European countries in our analysis can be categorized into countries that had an early and strong response (Italy, France, Spain) and those that had a more moderate response (Germany, Netherlands, Belgium, United Kingdom) [34]. For example, Italy experienced a large COVID-19 outbreak early on in March 2020, prompting early public health interventions in early March. Similar measures were implemented in mid-March in Spain and France [33]. Mandatory masks were introduced relatively early, by mid-March in France and late April in Italy. Some restrictions were relaxed during the summer, depending on the pandemic trends, but then re-imposed when the pandemic worsened in the fall [34]. By contrast, Germany, the Netherlands, Belgium and the United Kingdom had a more moderate response, characterized by a more gradual implementation of social distancing and mask-wearing measures [33]. Movement restrictions were implemented by late March, but mask mandates varied. For example, the Netherlands only introduced masks on public transport in June. In general, the implementation of public health interventions also varied by subnational region (e.g., in Germany movement restrictions were only implemented at regional level, not nationwide) [34].

Relating this country classification to our results, we do not observe any systematic pattern linking countries with stronger or more moderate responses to specific advertisement image effects. For instance, while the association between the "woman blowing nose" image (image 3) and higher threat perception to oneself in Italy could suggest increased awareness of symptomatic COVID-19 indicators, this effect is not observed in France or Spain. Similarly, within the group of countries classified as having a moderate response, contrasting image effects emerge. For example, the "couple blowing nose" image (image 4) was associated with higher mask-wearing in Germany but lower in the United Kingdom.

Additionally, we observe similar imagery effects across countries with different COVID-19 policy strategies. For example, the "couple blowing nose" image (image 4) was associated with a lower likelihood of increased handwashing in both Italy and the United Kingdom. In the United States, where COVID-19 quickly became a highly polarizing and political issue [35,36], this polarization may have amplified self-selection effects, particularly for advertising images explicitly referring to COVID-19 or mask-wearing. It is important to note that these interpretations, in relation to country-specific policies, should be approached with caution. While country-specific policies offer additional context for our results, further analysis would be needed to empirically verify potential associations between COVID-19 policy and advertisement design. While this is beyond the focus of this work, it highlights the importance of future survey research using Facebook advertisements during (health) crises to carefully consider the local political environment and public sentiment when designing ads, but also that image effects may be influenced by factors beyond the crisis itself. An important consideration is that our findings may not generalize beyond Facebook, as platform-specific algorithms with targeting mechanisms unknown to researchers may influence self-selection bias differently across social media platforms. We discuss additional limitations of our study in the next section.

## Limitations

This study is subject to several limitations pertaining to the design of the survey, the configuration of the Meta advertisement system, and the differences in the COVID-19 pandemic and public health interventions at the time across countries.

First of all, it is important to note that our survey data collection began at different times, with variations of up to three weeks across the countries, ranging from March 13 to April 2, 2020. Analyzing the images' performance over time, we observed that during the first weeks of the advertising campaigns, the images with the highest topic salience (i.e., the mask-wearing images) recruited the majority of respondents. However, the self-selection bias arising from interest in the survey topic might have been less pronounced than expected. This could be attributed to the various non-pharmaceutical interventions and lockdown measures in effect during that period, which likely led people to spend more time at home and, thus, more time online and on Facebook. This may limit the generalizability of our findings to other public health topics or to contexts with less urgent health threats. Moreover, although some of our advertisements did not explicitly mention the survey topic of the COVID-19 pandemic, they still referred to health-related topics in the advertisement text and in the name of our Facebook project page (i.e., "Health Behavior Survey"). This reference to health-related topics in the advertisements might have offset some of the effects attributed to the visual image, potentially attracting respondents with higher concerns or interest in health topics. However, previous research contradicts this assumption, indicating that the visual image in an advertisement has a more substantial impact than the accompanying text [19]. We lacked an independent measure of respondents' pre-survey interest in the survey topic. As a result, we could only infer that the advertisement images making the survey topic more salient attracted respondents with a higher level of interest in it. Since respondents were recruited during the early months of the pandemic, it is reasonable to assume a heightened general interest in COVID-19. This may have led to an underestimation of the role of the advertisement image in self-selection bias, as there was no direct measure of respondents' interest in the survey topic. Future studies could address this issue by including a question about respondents' interest in the survey topic or motivation to participate. When using ad images with different topics, researchers could ask about respondents' interest in all topics mentioned in the ads—since

they only know which ad image respondents entered after data collection—or design separate surveys with unique links for each ad image, explicitly measuring interest in the displayed topic. In addition, the A/B testing feature in the campaign setup could be used to randomize the display of images.

As the Meta advertisement system is a business tool, repurposed for research, there are drawbacks that come with it. The optimization towards the campaign objective, here "traffic", means that we may face the risk of receiving an unbalanced sample in terms of characteristics that are associated with clicking on a survey link, such as sex, age or education. However, we employed demographic targeting as a measure to mitigate the effects of this allocation to specific user groups [25]. When multiple ads with different images are used within the same advertisement set, as was the case in this study, the optimization algorithm will, over time, show those images more frequently that drive more link clicks [26]. This leads to advertisement images in the advertisement set being presented at a different rate to the target audience, with a preference towards those advertisements which will results in the lowest cost per click for the specific demographic, by age and sex. In our case this means that the athlete images (image 1 and 2) were displayed less and the high topic salience images (image 5 and 6) were shown to more users, which is reflected in our distribution of completed surveys by image. In our regression analysis we adjust for the difference in sample composition between the images. Thus, for the comparison of the outcomes of threat perception of COVID-19 and preventive behavior between the images, we reduce the variations in sample composition that may arise due to differential exposure of each image to the target audience. However, it is important to acknowledge that the ad optimization algorithm remains opaque to researchers outside of Meta. Future survey recruitment campaigns could achieve a more controlled display of the different images by creating one campaign per image, as it was done by Zindel [37]. Further, we need to acknowledge that the results between social media platforms cannot be directly compared as the advertisement structure and platform usage pattern may differ between them. However, as the majority of studies that use social media as a recruitment tool rely on Facebook, the results presented here still offer valuable insights for those studies using Facebook early during the pandemic.

Despite these limitations, our study provides a significant contribution to the existing literature. Here we shifted the conventional attention on participation rates to focus the analysis of survey responses influenced by advertisement content. Furthermore, our research offers insights drawn from a cross-national analysis encompassing eight different countries in Europe and North America. Lastly, our emphasis lies in capturing trends within the general online population, which allows for broader applicability in the context of population-wide public health crises, such as pandemics or natural disasters, rather focusing on specific subpopulations — such as parents, smokers, or men — as previous research has done.

## Conclusion

Our findings provide limited support for our initial hypothesis that, in the context of Facebook-based recruitment, higher topic salience in the survey recruitment materials would lead to systematically different survey responses. However, within specific countries, we observed that certain recruitment images were associated with different responses regarding perceived COVID-19 risk perception and preventive behaviors. The direction of these effects varied by outcome and country, suggesting that image effects are context-specific and do not uniformly bias responses. It is important to note that our results are specific to the unique circumstances of the COVID-19 pandemic. For future applications of survey recruitment via Facebook advertisements during (health) crises, we conclude that thematic

advertisements can effectively reach a broad audience without consistently introducing systematic biases. However, while salient images may be useful for broad engagement, we recommend that researchers remain mindful of potential contextual biases and consider pre-testing recruitment materials, where feasible. Future research should explore whether this effect holds for other social media platforms, non-crisis periods, and different, potentially polarizing survey topics. In conclusion, while topical self-selection had a limited overall impact, context-specific image effects suggest that careful recruitment design remains important for Facebook recruited surveys during health crises.

For more information, see S1 File.

## Supporting information

**S1 File. Supporting information.**
(PDF)

**S2 File. Supporting information: Regression Tables.**
(PDF)

**S3 File. Supporting information: Analysis Code.**
(ZIP)

**S4 File. Supporting Information: Aggregated analysis data.**
(ZIP)

## Acknowledgments

We thank Dr. André Grow-Böser for his work on the study design and data collection, and for devising the initial idea for this analysis. Jessica Donzowa gratefully acknowledges the resources provided by the International Max Planck Research School for Population, Health and Data Science (IMPRS-PHDS).

## Author contributions

**Conceptualization:** Jessica Donzowa, Daniela Perrotta, Emilio Zagheni.

**Data curation:** Jessica Donzowa, Daniela Perrotta.

**Formal analysis:** Jessica Donzowa.

**Funding acquisition:** Emilio Zagheni.

**Investigation:** Jessica Donzowa, Daniela Perrotta, Emilio Zagheni.

**Methodology:** Jessica Donzowa.

**Project administration:** Jessica Donzowa.

**Resources:** Daniela Perrotta, Emilio Zagheni.

**Software:** Jessica Donzowa.

**Supervision:** Daniela Perrotta, Emilio Zagheni.

**Visualization:** Jessica Donzowa, Daniela Perrotta.

**Writing – original draft:** Jessica Donzowa.

**Writing – review & editing:** Jessica Donzowa, Daniela Perrotta, Emilio Zagheni.

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
