## [Decision Letter · Decision Letter 0]

29 Sep 2024

PONE-D-24-35721Assessing self-selection biases in online surveys: Evidence from the COVID-19 Health Behavior SurveyPLOS ONE

Dear Dr. Donzowa,

Thank you for submitting your manuscript to PLOS ONE. After careful consideration, we feel that it has merit but does not fully meet PLOS ONE’s publication criteria as it currently stands. Therefore, we invite you to submit a revised version of the manuscript that addresses the points raised during the review process.

We look forward to receiving your revised manuscript.

Kind regards,

Anat Gesser-Edelsburg, Ph.D.

Academic Editor

PLOS ONE

Journal Requirements: When submitting your revision, we need you to address these additional requirements. 1. Please ensure that your manuscript meets PLOS ONE's style requirements, including those for file naming. The PLOS ONE style templates can be found at https://journals.plos.org/plosone/s/file?id=wjVg/PLOSOne_formatting_sample_main_body.pdf and https://journals.plos.org/plosone/s/file?id=ba62/PLOSOne_formatting_sample_title_authors_affiliations.pdf 2. Thank you for stating the following financial disclosure: "This study was funded with support from the Max Planck Institute for Demographic Research, which is part of the Max Planck Society. " Please state what role the funders took in the study.  If the funders had no role, please state: ""The funders had no role in study design, data collection and analysis, decision to publish, or preparation of the manuscript."" If this statement is not correct you must amend it as needed. Please include this amended Role of Funder statement in your cover letter; we will change the online submission form on your behalf. 3. In the online submission form, you indicated that "Interested researchers wishing to conduct scientific research can access the survey data upon request and approval. Contact details for accessing the data: Perrotta, Daniela, perrotta@demogr.mpg.de." All PLOS journals now require all data underlying the findings described in their manuscript to be freely available to other researchers, either 1. In a public repository, 2. Within the manuscript itself, or 3. Uploaded as supplementary information.This policy applies to all data except where public deposition would breach compliance with the protocol approved by your research ethics board. If your data cannot be made publicly available for ethical or legal reasons (e.g., public availability would compromise patient privacy), please explain your reasons on resubmission and your exemption request will be escalated for approval. 4. When completing the data availability statement of the submission form, you indicated that you will make your data available on acceptance. We strongly recommend all authors decide on a data sharing plan before acceptance, as the process can be lengthy and hold up publication timelines. Please note that, though access restrictions are acceptable now, your entire data will need to be made freely accessible if your manuscript is accepted for publication. This policy applies to all data except where public deposition would breach compliance with the protocol approved by your research ethics board. If you are unable to adhere to our open data policy, please kindly revise your statement to explain your reasoning and we will seek the editor's input on an exemption. Please be assured that, once you have provided your new statement, the assessment of your exemption will not hold up the peer review process. 5. We note that Figure 1 includes an image of a [patient / participant / in the study].  As per the PLOS ONE policy (http://journals.plos.org/plosone/s/submission-guidelines#loc-human-subjects-research) on papers that include identifying, or potentially identifying, information, the individual(s) or parent(s)/guardian(s) must be informed of the terms of the PLOS open-access (CC-BY) license and provide specific permission for publication of these details under the terms of this license. Please download the Consent Form for Publication in a PLOS Journal (http://journals.plos.org/plosone/s/file?id=8ce6/plos-consent-form-english.pdf). The signed consent form should not be submitted with the manuscript, but should be securely filed in the individual's case notes. Please amend the methods section and ethics statement of the manuscript to explicitly state that the patient/participant has provided consent for publication: “The individual in this manuscript has given written informed consent (as outlined in PLOS consent form) to publish these case details”.  If you are unable to obtain consent from the subject of the photograph, you will need to remove the figure and any other textual identifying information or case descriptions for this individual.

Reviewers' comments:

Reviewer's Responses to Questions

**Comments to the Author**

1. Is the manuscript technically sound, and do the data support the conclusions?

Reviewer #1: Yes

Reviewer #2: Yes

2. Has the statistical analysis been performed appropriately and rigorously? 

Reviewer #1: Yes

Reviewer #2: Yes

3. Have the authors made all data underlying the findings in their manuscript fully available?

Reviewer #1: No

Reviewer #2: Yes

4. Is the manuscript presented in an intelligible fashion and written in standard English?

Reviewer #1: Yes

Reviewer #2: Yes

5. Review Comments to the Author

Reviewer #1: This is a great manuscript. Please consider my questions and suggestions to improve and clarify results. My primary concerns are around the conclusions and interpretation of results. You say multiple times that image salience has no effect, however throughout the results, it is pointed out multiple times that images do in fact influence the resulting sample's perceived risk and behaviors. It would also be important to see an overall pooled analysis of all countries. Why was this not done? Further, I suspect there may be effect modification by month of pandemic, so highly recommend exploring whether there is image x month interaction in regression models.

Reviewer #2: PONE-D-24-35721

Assessing self-selection biases in online surveys: Evidence from the COVID-19 Health

Behavior Survey

Reviewer comments

Summary

Overall, I thought the manuscript focus and hypotheses were quite interesting with regards to leverage-salience theory applications to COVID-19. This topic seems quite relevant, given the need to better understand disease outcomes and population perceptions. From a constructively critical standpoint, I did find the manuscript results a bit hard to follow. I think the authors could attempt to lay out the results more succinctly. I also think the visualization of the results could be more effectively presented. In your appendix 1, your S1 figure (number of participants by ad image and country over time) is a great depiction of the data and might be a nice addition to the actual manuscript (potentially at the beginning of your results?).

I would similarly think about how to better represent model significance. As a reader, I would really like to be able to capture that information visually, or in some way that allows me easily to see how some factors (i.e. month of survey participation) were impactful on model outcomes.

Introduction

Well structured. However, near the end of your introduction, you start to discuss the results of your analysis. I might remove that, and instead, formulate some questions that relate to your hypotheses. Do advertising images have a variable effect on survey outcomes? What factors (location, timeframe, population demographics, etc) might influence survey outcomes, given the heterogenous nature of leverage and salience?

I might also broaden the hypothesis to say that there are several influential factors which may determine who participates in the survey – which includes time frame of the pandemic – but also includes many other factors, and your intent is to tease out these interactions and effects based on the multi-level structure you have constructed (age/sex/location etc.).

Methods

I can appreciate the detail in the methodology, which is well organized. I may be a little fixated on visualizing outcomes, but you might consider one diagram which depicts the flow of your methodology – even if its in a supplemental appendix. Your approach focuses on survey data from this 2020 time frame, based on 6 differing ads which have some form of increasing COVID-19 prominence. Then you analyze based on several predictor factors. That could be presented in one diagram – so the reader would immediately be able to grasp the approach and how the models are applied.

Also – you mention a total of 32 models, I’m assuming based on the changing factor levels. If that is clearly defined, maybe I just missed it – but if not, id include that. Potentially in the diagram that I’m suggesting above.

Results

Page 9: “Fig 2 shows the variation in the levels of perceived threat towards COVID-19 that respondents reported for themselves and for their family, across the various countries and advertisement images. Respondents generally perceived COVID-19 as a higher threat to their family than to themselves.”

I couldn’t tell from Figure 2 that respondents perceived COVID-19 as a higher threat to their family than themselves? Am I missing something?

Also – if your models are binomial and multinomial logistic regressions, then a prominent summary statistic would be the odds ratios. I see you are reporting these in your S2 appendix. Maybe summarizing the odds ratios for the most influential models might be interesting to report in the manuscript? Again, if I missed that in the manuscript, my apologies. You could also present a standardized coefficient diagram that indicates those factors which are significant, with their odds ratios/Cis. Maybe that’s already in the supplementals. If so – one of those diagrams in the results would seem to be very useful.

Discussion

I think your discussion would benefit by giving some thought as to WHY you think some of this variability exists – either across location or the differing interactions of influential factors on each of the images. My sense is – some of this language is more appropriate for the results section.

For example: you report some of the locational differences (based on country). Were their known policy or covid-19 spread variations based on county – that might be associated with what your results are indicating? I realize that’s not the focus of this work – but I’m interested in why we are seeing these results, and what implications we might be infer?

6. PLOS authors have the option to publish the peer review history of their article (what does this mean?). If published, this will include your full peer review and any attached files.

Reviewer #1: No

Reviewer #2: No

---

## [Author Response · Author response to Decision Letter 1]

17 Dec 2024

Editor Comments:

Thank you for your reminder regarding PLOS ONE's style requirements. We have carefully reviewed and followed the formal guidelines, including file naming conventions. However, if we have misunderstood any of the requirements, we would be happy to make any additional adjustments.

2. Thank you for stating the following financial disclosure: "This study was funded with support from the Max Planck Institute for Demographic Research, which is part of the Max Planck Society. "

If this statement is not correct you must amend it as needed. Please include this amended Role of Funder statement in your cover letter; we will change the online submission form on your behalf.

We have amended the Role of Funder statement as follows:

“This study was funded with support from the Max Planck Institute for Demographic Research, which is part of the Max Planck Society. The funders had no role in study design, data collection and analysis, decision to publish, or preparation of the manuscript.”

3. In the online submission form, you indicated that "Interested researchers wishing to conduct scientific research can access the survey data upon request and approval. Contact details for accessing the data: Perrotta, Daniela, perrotta@demogr.mpg.de."

Due to privacy regulations and the potential risk of identifying respondents, we cannot share the raw individual survey data used in our analysis. However, we will make the data available upon request to researchers for the purpose of scientific research. Additionally, in the revised version of the manuscript, we have now included the aggregated data underlying the main findings as supplementary material.

As mentioned in the previous point, we cannot share the individual raw survey data due to data protection regulations. However, in the revised version of the manuscript, we have included the aggregated data underlying the graphs, as well as the code used for the analysis, which allows reproduction of the figures presented in the paper.

5. We note that Figure 1 includes an image of a [patient / participant / in the study].

Figure 1 in the manuscript shows the six images used in the advertisements placed on Facebook to recruit survey participants. The images were purchased from two stock image platforms, Adobe Stock and iStockphoto, under licenses permitting republication. The individuals depicted in the stock image were not participants in our study. Consequently, we do not possess or disclose any identifying or potentially identifying information about them.

Reviewer(s)' Comments to Author:

Reviewer #1: This is a great manuscript. Please consider my questions and suggestions to improve and clarify results. My primary concerns are around the conclusions and interpretation of results. You say multiple times that image salience has no effect, however throughout the results, it is pointed out multiple times that images do in fact influence the resulting sample's perceived risk and behaviors. It would also be important to see an overall pooled analysis of all countries. Why was this not done? Further, I suspect there may be effect modification by month of pandemic, so highly recommend exploring whether there is image x month interaction in regression models.

We would like to thank the reviewer for their appreciation of our manuscript.

We appreciate the reviewer raising these important points regarding our conclusion and interpretation of our results. As stated in the response to comment 12 in the response letter we have revised our conclusion to provide a more nuanced interpretation of our findings (lines 568 to 584 in the revised manuscript). We have also revised the discussion section to interpret our findings in the context of differing public health interventions and the progression of the COVID-19 pandemic in each country. In the revised version of the manuscript, we highlight this aspect in the Discussion section on the lines 466 to 508.

In response to the reviewer comment, we have computed an overall pooled analysis across all countries, controlling for image, gender, age, education, month of survey participation, and country. Both the pooled and country-specific analyses show the same general pattern: the adoption of hand washing was higher than the adoption of wearing a face mask, and the perceived threat to family was higher than the perceived threat to self. However, the pooled analysis overlooks variations in image effects across countries. Therefore, we believe that the country-specific results are preferable as they highlight the varying effect of the images by country. We explain this in detail in our response to comment 14 in the response letter.

We agree with the reviewer that exploring the interaction between the month of survey participation and the image effect is relevant to our research question. In response, we have tested whether there is an interaction between the ad image and the month of survey participation by including an interaction term in our models. However, upon inspection, we found that including the interaction terms resulted in inflated standard errors and overly wide confidence intervals. This is due to small case numbers or empty cells in the crosstabulation between our outcome variables (image and month), resulting in sparse data for some combinations of the interaction terms. As a result, our sample size does not allow us to calculate this interaction with adequate precision. Given this limitation, we decided not to include this estimation in the manuscript. In the revised version of the manuscript, we now mention this in lines 405 to 409 of the Results section. We refer to this in our response to comment 9 in the response letter.

Reviewer#2: PONE-D-24-35721

Assessing self-selection biases in online surveys: Evidence from the COVID-19 Health

Behavior Survey

Reviewer comments

Summary

Overall, I thought the manuscript focus and hypotheses were quite interesting with regards to leverage-salience theory applications to COVID-19. This topic seems quite relevant, given the need to better understand disease outcomes and population perceptions. From a constructively critical standpoint, I did find the manuscript results a bit hard to follow. I think the authors could attempt to lay out the results more succinctly. I also think the visualization of the results could be more effectively presented. In your appendix 1, your S1 figure (number of participants by ad image and country over time) is a great depiction of the data and might be a nice addition to the actual manuscript (potentially at the beginning of your results?).

We appreciate the reviewer’s recognition of the relevance of our research topic and theoretical framing. In response, we have revised the Results section of our manuscript to improve readability. Specifically, we ensured that the results for each outcome are consistently presented and also streamlined our phrasing to avoid complex or contradictory expressions. Please see our response to comment 2 in the response letter for a detailed description of the changes implemented to clarify the presentation of our results.Additionally, we have improved the readability of Figure 2 (now Figure 3 in the revised manuscript) by presenting the results as a heat map instead of a stacked bar plot and improved the visualization of significant differences in the model plots (Figure 5 and 6 in the revised manuscript) by adding a vertical line corresponding to the lowest confidence interval (CI). This provides a clearer visual cue to identify overlapping confidence intervals and enhances the reader's ability to assess significant differences between the images. We thank the reviewer for their appreciation of our visualization of the S1 figure. In response, we now include it as Figure 2 in the Results section of the revised manuscript, where we describe the number of participants by ad image over time (lines 281-282).

I would similarly think about how to better represent model significance. As a reader, I would really like to be able to capture that information visually, or in some way that allows me easily to see how some factors (i.e. month of survey participation) were impactful on model outcomes.

We appreciate the reviewer’s suggestion to visually capture the impact of factors, such as the month of survey participation, on model outcomes. As noted in our detailed response to comments 9 and 10 in the response letter, we carefully considered reporting the stepwise odds ratios or a standardized coefficient diagrams in the main text. However, we concluded that reporting predicted probabilities for the final model in the main text provides a clearer and more direct comparison across categories, aligning better with our study's objectives. We reported the predicted probabilities for each model step in Appendix 1 of the original manuscript, which we also reference in the main text when describing the explanatory effect of the covariates. We considered including these figures in the main text, but we chose to place them in the Appendix to avoid overloading the manuscript with additional figures. For a more detailed explanation, please refer to our response to comments 9 and 10 in the response letter.

Introduction

Well structured. However, near the end of your introduction, you start to discuss the results of your analysis. I might remove that, and instead, formulate some questions that relate to your hypotheses. Do advertising images have a variable effect on survey outcomes? What factors (location, timeframe, population demographics, etc) might influence survey outcomes, given the heterogenous nature of leverage and salience? I might also broaden the hypothesis to say that there are several influential factors which may determine who participates in the survey – which includes time frame of the pandemic – but also includes many other factors, and your intent is to tease out these interactions and effects based on the multi-level structure you have constructed (age/sex/location etc.).

We are pleased to read that the reviewer finds our introduction well structured. Following the reviewer’s suggestion, we have edited the end of the Introduction section (lines 62-71) accordingly.

Methods

I can appreciate the detail in the methodology, which is well organized. I may be a little fixated on visualizing outcomes, but you might consider one diagram which depicts the flow of your methodology – even if its in a supplemental appendix. Your approach focuses on survey data from this 2020 time frame, based on 6 differing ads which have some form of increasing COVID-19 prominence. Then you analyze based on several predictor factors. That could be presented in one diagram – so the reader would immediately be able to grasp the approach and how the models are applied.Also – you mention a total of 32 models, I’m assuming based on the changing factor levels. If that is clearly defined, maybe I just missed it – but if not, id include that. Potentially in the diagram that I’m suggesting above.

We thank the reviewer for their appreciation of the detail and structure of our methodology and the suggestion to include a visualization of our study design. In response, we have created a figure that visualizes our study design and included it in the supplementary material (S1 Fig). We reference this figure at Line 218 of the revised version of the manuscript. We apologize that the total number of our models was not clear. The total of 32 models results from calculating models for four outcomes across eight countries, representing the number of final models that include all covariates. In the revised version of the manuscript, we have clarified this point in the Materials and methods section at line 220.

Results

Page 9: “Fig 2 shows the variation in the levels of perceived threat towards COVID-19 that respondents reported for themselves and for their family, across the various countries and advertisement images. Respondents generally perceived COVID-19 as a higher threat to their family than to themselves.”I couldn’t tell from Figure 2 that respondents perceived COVID-19 as a higher threat to their family than themselves? Am I missing something?

Following the reviewer’s comment, we have improved Figure 2 (now Figure 3 in revised manuscript) for better readability and now report the level of threat perception by image in the form of a heat map instead of a stacked bar plot.

Also – if your models are binomial and multinomial logistic regressions, then a prominent summary statistic would be the odds ratios. I see you are reporting these in your S2 appendix. Maybe summarizing the odds ratios for the most influential models might be interesting to report in the manuscript? Again, if I missed that in the manuscript, my apologies. You could also present a standardized coefficient diagram

---

## [Decision Letter · Decision Letter 1]

24 Feb 2025

PONE-D-24-35721R1Assessing self-selection biases in online surveys: Evidence from the COVID-19 Health Behavior SurveyPLOS ONE

Dear Dr. Donzowa,

Thank you for submitting your manuscript to PLOS ONE. After careful consideration, we feel that it has merit but does not fully meet PLOS ONE’s publication criteria as it currently stands. Therefore, we invite you to submit a revised version of the manuscript that addresses the points raised during the review process.

We look forward to receiving your revised manuscript.

Kind regards,

Prof. Anat Gesser-Edelsburg, Ph.D.

Academic Editor

PLOS ONE

Journal Requirements:

Reviewers' comments:

Reviewer's Responses to Questions

**Comments to the Author**

1. If the authors have adequately addressed your comments raised in a previous round of review and you feel that this manuscript is now acceptable for publication, you may indicate that here to bypass the “Comments to the Author” section, enter your conflict of interest statement in the “Confidential to Editor” section, and submit your "Accept" recommendation.

Reviewer #2: All comments have been addressed

Reviewer #3: (No Response)

2. Is the manuscript technically sound, and do the data support the conclusions?

Reviewer #2: Yes

Reviewer #3: Yes

3. Has the statistical analysis been performed appropriately and rigorously? 

Reviewer #2: Yes

Reviewer #3: Yes

4. Have the authors made all data underlying the findings in their manuscript fully available?

Reviewer #2: Yes

Reviewer #3: No

5. Is the manuscript presented in an intelligible fashion and written in standard English?

Reviewer #2: Yes

Reviewer #3: Yes

6. Review Comments to the Author

Reviewer #2: The authors have complied with all the requested revisions and feedback. Recommend acceptance of the manuscript.

Reviewer #3: This study investigates whether self-selection bias affects responses in online surveys when participants are recruited through Facebook advertisements with varying degrees of topic salience. Using a large cross-national dataset (N=120,184) from eight countries, the authors find minimal self-selection bias, suggesting that concerns over bias in social media-recruited surveys may be overstated. The study is methodologically rigorous, well-written, and highly relevant to discussions on the validity of online survey data.

However, while the paper makes a strong contribution, there are some limitations and areas for improvement that should be addressed before publication. Below, I provide specific comments on the title, strengths, and areas for revision.

Major Comment: Title Revision Needed

Clarify That Findings Are Facebook-Specific

• The current title, Assessing Self-Selection Biases in Online Surveys, is too broad given that all recruitment was conducted via Facebook ads.

• Since self-selection bias may differ across platforms (e.g., LinkedIn, Twitter, Instagram), the title should explicitly mention Facebook to avoid overgeneralization.

• Suggested revision: Assessing Self-Selection Biases in Facebook-Recruited Online Surveys: Evidence from the COVID-19 Health Behavior Survey

• Alternatively, the authors could add a phrase in the abstract emphasizing that findings may not generalize to other platforms.

Strengths:

1. Timely and Important Topic

o The study addresses a key methodological concern in online survey research—whether recruitment via social media advertisements introduces selection bias.

o Given the widespread use of online surveys, particularly during and after the COVID-19 pandemic, this research is highly relevant.

2. Large and Cross-National Dataset

o The inclusion of 120,184 respondents from eight countries enhances the generalizability of the findings.

o Few studies analyze cross-national differences in self-selection bias, making this study a valuable contribution.

3. Innovative Use of Ad Image Variations

o The quasi-experimental design, which compares survey responses from participants recruited through different advertisement images, is an innovative approach to testing self-selection bias.

o This provides empirical evidence on how survey topic salience in recruitment ads affects participation and response patterns.

4. Strong Statistical Methods

o The study uses multinomial and binomial logistic regression models, effectively controlling for demographics (age, gender, education), country context, and timing of survey participation.

o The use of stepwise regression models to test for the persistence of self-selection effects adds credibility to the conclusions.

5. Important Practical Implications

o The study challenges the common assumption that social media-based surveys suffer from strong self-selection bias.

o This finding is particularly useful for researchers designing future online surveys and suggests that Facebook-based survey recruitment remains a viable method.

Areas for Improvement & Revisions:

1. Limited Generalizability Beyond Facebook

o The authors should explicitly clarify that findings may not generalize to other social media platforms with different ad targeting mechanisms.

o Recommendation: Add a paragraph in the discussion section acknowledging that platform algorithms may vary, affecting how self-selection bias manifests elsewhere.

2. Lack of Pre-Survey Interest Measurement

o The authors assume that ad images influenced self-selection bias, but they do not measure participants’ pre-existing interest in COVID-19 topics.

o Would it be possible to use post-hoc measures (e.g., interest in public health topics) to strengthen the interpretation of self-selection effects?

o At minimum, the authors should discuss this limitation and suggest how future studies might address it.

3. National Contexts & Policy Differences Not Fully Accounted For

o COVID-19 policies and public health messaging varied significantly across the eight countries in the study.

o Did differences in government mandates, media coverage, or pandemic severity influence how participants perceived survey topics?

o The authors should expand their discussion on how these external factors might have affected responses.

4. Potential Algorithm Bias in Ad Distribution

o The paper notes that Facebook’s algorithm optimized for engagement, meaning some ad images were shown more frequently than others.

o However, it is unclear how much this influenced sample composition.

o Would it be possible to report how often each ad image was shown across demographic groups?

o If this data is unavailable, the authors should discuss the potential impact of algorithmic bias in the limitations section.

5. Unclear Impact on Long-Term Online Survey Practices

o While the findings suggest that self-selection bias is minimal in social media-recruited surveys, does this hold outside of a crisis situation like COVID-19?

o The authors should consider whether the unique context of the pandemic (increased time online, heightened public health awareness) might have reduced typical self-selection effects.

o Future research directions should explore non-crisis topics and different recruitment methods.

Final Recommendation:

This study makes a strong and novel contribution to the field of online survey methodology. I recommend acceptance with minor revisions to strengthen the title, discussion on generalizability, and analysis of potential biases. With these refinements, the paper will be even more impactful for researchers designing future online surveys.

Would the authors be open to modifying the title to specify Facebook recruitment? Additionally, addressing algorithmic bias and survey interest measurement would make the findings more robust.

I appreciate the opportunity to review this manuscript and look forward to seeing the final version.

Best regards,

7. PLOS authors have the option to publish the peer review history of their article (what does this mean?). If published, this will include your full peer review and any attached files.

Reviewer #2: **Yes: **Erich Seamon

Reviewer #3: No

---

## [Author Response · Author response to Decision Letter 2]

16 Apr 2025

Reviewer(s)' Comments to Author:

Reviewer 2:

The authors have complied with all the requested revisions and feedback. Recommend acceptance of the manuscript.

We thank the reviewer for the positive feedback and recommendation to accept our manuscript. We greatly appreciate their support throughout the review process.

Reviewer 3:

This study investigates whether self-selection bias affects responses in online surveys when participants are recruited through Facebook advertisements with varying degrees of topic salience. Using a large cross-national dataset (N=120,184) from eight countries, the authors find minimal self-selection bias, suggesting that concerns over bias in social media-recruited surveys may be overstated. The study is methodologically rigorous, well-written, and highly relevant to discussions on the validity of online survey data. However, while the paper makes a strong contribution, there are some limitations and areas for improvement that should be addressed before publication. Below, I provide specific comments on the title, strengths, and areas for revision.

We thank the reviewer for their thoughtful and encouraging feedback. We appreciate their positive assessment of our study’s methodological rigor, writing quality, and contribution to discussions on online survey quality. The additional points raised by the reviewer provided valuable insights that helped us further refine and improve our manuscript. We addressed the specific points in the responses below and incorporated the suggestions into the revised version of the manuscript.

Major Comment: Title Revision Needed

Clarify That Findings Are Facebook-Specific

● The current title, Assessing Self-Selection Biases in Online Surveys, is too broad given that all recruitment was conducted via Facebook ads.

● Since self-selection bias may differ across platforms (e.g., LinkedIn, Twitter, Instagram), the title should explicitly mention Facebook to avoid overgeneralization.

● Suggested revision: Assessing Self-Selection Biases in Facebook-Recruited Online Surveys: Evidence from the COVID-19 Health Behavior Survey

● Alternatively, the authors could add a phrase in the abstract emphasizing that findings may not generalize to other platforms.

We thank the reviewer for raising this point. We agree that it is important to address the platform-specific nature of our findings from social media-recruited surveys in general, and to make this explicit in the title. We agree with the title suggested by the reviewer, “Assessing self-selection biases in Facebook-recruited online surveys: Evidence from the COVID-19 Health Behavior Survey”, and incorporated this change in the revised version of our manuscript.

Strengths:

1. Timely and Important Topic

● The study addresses a key methodological concern in online survey research—whether recruitment via social media advertisements introduces selection bias.

● Given the widespread use of online surveys, particularly during and after the COVID-19 pandemic, this research is highly relevant.

2. Large and Cross-National Dataset

● The inclusion of 120,184 respondents from eight countries enhances the generalizability of the findings.

● Few studies analyze cross-national differences in self-selection bias, making this study a valuable contribution.

3. Innovative Use of Ad Image Variations

● The quasi-experimental design, which compares survey responses from participants recruited through different advertisement images, is an innovative approach to testing self-selection bias.

● This provides empirical evidence on how survey topic salience in recruitment ads affects participation and response patterns.

4. Strong Statistical Methods

● The study uses multinomial and binomial logistic regression models, effectively controlling for demographics (age, gender, education), country context, and timing of survey participation.

● The use of stepwise regression models to test for the persistence of self-selection effects adds credibility to the conclusions.

5. Important Practical Implications

● The study challenges the common assumption that social media-based surveys suffer from strong self-selection bias.

● This finding is particularly useful for researchers designing future online surveys and suggests that Facebook-based survey recruitment remains a viable method.

We thank the reviewer for their thoughtful feedback on our manuscript and for pointing out the strengths of our study, including its relevance and timeliness, the use of a large cross-national dataset, the innovative use of ad image variations and appropriate statistical methods, and its important practical implications for survey research.

Areas for Improvement & Revisions:

1. Limited Generalizability Beyond Facebook

● The authors should explicitly clarify that findings may not generalize to other social media platforms with different ad targeting mechanisms.

● Recommendation: Add a paragraph in the discussion section acknowledging that platform algorithms may vary, affecting how self-selection bias manifests elsewhere.

We thank the reviewer for pointing out the important influence of platform-specific algorithms on social media-based research. We agree that it is important to acknowledge the challenges and limitations of generalizing findings beyond a specific social media platform due to the unknown mechanisms in the ads distribution.

In the original version of this manuscript, we mentioned this aspect in the Limitations section (lines 568 - 572):

“Further, we need to acknowledge that the results between social media platforms cannot be directly compared as the advertisement structure and platform usage pattern may differ between them. However, as the majority of studies that use social media as a recruitment tool rely on Facebook, the results presented here still offer valuable insights for those studies using Facebook early during the pandemic.”

To address this aspect more explicitly, we have also added a sentence in the Discussion section highlighting that the results may not be generalizable beyond the Facebook platform. Lines 510 through 514 of the revised manuscript now read:

“An important consideration is that our findings may not generalize beyond Facebook, as platform-specific algorithms with targeting mechanisms unknown to researchers may influence self-selection bias differently across social media platforms. We discuss additional limitations of our study in the next section.”

2. Lack of Pre-Survey Interest Measurement

● The authors assume that ad images influenced self-selection bias, but they do not measure participants’ pre-existing interest in COVID-19 topics.

● Would it be possible to use post-hoc measures (e.g., interest in public health topics) to strengthen the interpretation of self-selection effects?

● At minimum, the authors should discuss this limitation and suggest how future studies might address it.

We appreciate the reviewer's point about the limitation of not measuring participants' pre-existing interest in the COVID-19 topics. In the Limitations section of the original manuscript (lines 535-538), we acknowledged the lack of an independent measure of respondents' initial interest in the survey topic. In response to the reviewer's comment, we have expanded this section to discuss the potential implications for interpreting our results and suggest ways future studies might address this issue.

The revised version of the manuscript now reads (lines 538-547):

"Since respondents were recruited during the early months of the pandemic, it is reasonable to assume a heightened general interest in COVID-19. This may have led to an underestimation of the role of the advertisement image in self-selection bias, as there was no direct measure of respondents' interest in the survey topic. Future studies could address this issue by including a question about respondents' interest in the survey topic. When using ad images with different topics, researchers could ask about respondents' interest in all topics mentioned in the ads—since they only know which ad image respondents entered after data collection—or design separate surveys with unique links for each ad image, explicitly measuring interest in the displayed topic."

3. National Contexts & Policy Differences Not Fully Accounted For

● COVID-19 policies and public health messaging varied significantly across the eight countries in the study.

● Did differences in government mandates, media coverage, or pandemic severity influence how participants perceived survey topics?

● The authors should expand their discussion on how these external factors might have affected responses.

We appreciate the reviewer’s comments regarding the potential influence of national contexts, such as government mandates, media coverage, and pandemic severity, on participants' perceptions of the survey topic. However, as noted in our response to comment two, we cannot make definitive claims about how these external factors shaped participants' perceptions, as we did not directly measure them.

In the original manuscript (lines 469-510), we had already provided an overview of the public health policies implemented across the eight countries in our study and analyzed their relationship to our findings. We observed no consistent pattern between image effects and countries, even when categorized by the stringency of their COVID-19 response measures during the initial months of the pandemic. One possible explanation is that individual-level factors—such as personal interest in COVID-19 or prior exposure to related information—may have played a more significant role than country-level policies in shaping participants’ engagement with the survey.

4. Potential Algorithm Bias in Ad Distribution

● The paper notes that Facebook’s algorithm optimized for engagement, meaning some ad images were shown more frequently than others.

● However, it is unclear how much this influenced sample composition.

● Would it be possible to report how often each ad image was shown across demographic groups?

● If this data is unavailable, the authors should discuss the potential impact of algorithmic bias in the limitations section.

We thank the reviewer for raising this point and suggesting an analysis of these metrics by demographic groups. In response, we added three additional tables to the S1 file (Table S5, S6, and S7) that show the number of impressions by gender, age, and advertisement design for the eight countries in our study. This additional information by age and gender improves the understanding of how the ads were displayed and provides insights into the age and gender composition of our sample by advertisement image. In the revised version of the manuscript, we now refer to these tables in the Results section (lines 290 to 295), where it reads:

“A more detailed description of the covariates by ad image, as well as impressions by age, gender and ad image, can be found in the supporting information S1 File (see S4 to S7 Tables in S1 File). In general, the age and gender composition by ad image is consistent with the number of impressions (i.e., the number of times the ad was displayed) recorded for the ad sets targeted by gender and age.”

We also added the main finding from these tables into the “Covariates by advertisement image” section of the S1 file, where we describe the sample composition by age, gender, and advertisement design. On page eight of the revised S1 file, it now reads:

“In general, the age and gender composition by ad image in each country aligns with the number of impressions (i.e., the number of times the ad was displayed) recorded for the ad sets targeting by gender and age (Tables S5 to S7). Taking the United States as an example, only 15% of the respondents recruited through the “group of athletes” image (image 2) reported being female in the survey, and the median age for this image is higher than the overall median age for the country in the survey–63 compared to 58. Looking at the number of impressions, we see that the “group of athletes” image (image 2) received a significantly higher share of impressions in the two older age groups (i.e., 45-65 and 65+) with about 54% compared to just 6% for the ad sets targeted to women in those same age groups.”

5. Unclear Impact on Long-Term Online Survey Practices

● While the findings suggest that self-selection bias is minimal in social media-recruited surveys, does this hold outside of a crisis situation like COVID-19?

● The authors should consider whether the unique context of the pandemic (increased time online, heightened public health awareness) might have reduced typical self-selection effects.

● Future research directions should explore non-crisis topics and different recruitment methods.

We thank the reviewer for raising the importance of the specific COVID-19 crisis context in which our survey was conducted. We agree that it is important to acknowledge that our results are specific to this health crisis and may not generalize to other potentially polarizing topics. We already emphasized this aspect in the original manuscript, for instance in the Conclusion section (line 590) where we limit the applicability of our results to “survey recruitment via Facebook advertisements during (health) crises”. In addition, we state the following in the Conclusion section of the original manuscript (lines 595-597):

“Future research should explore whether this effect holds for other social media platforms, non-crisis periods, and different, potentially polarizing survey topics.”

Final Recommendation:

This study makes a strong and novel contribution to the field of online survey methodology. I recommend acceptance with minor revisions to strengthen the title, discussion on generalizability, and analysis of potential biases. With these refinements, the paper will be even more impactful for researchers designing future online surveys.

Would the authors be open to modifying the title to specify Facebook recruitment? Additionally, addressing algorithmic bias and survey interest measurement would make the findings more robust.

I appreciate the opportunity to review this manuscript and look forward to seeing the final version.

We thank the reviewer for their thoughtful and supportive feedback. We appreciate the recognition of our study’s novel contribution to online survey methodology and the valuable suggestions for further strengthening the manuscript.

In response to the recommendations, we have revised the title to explicitly specify Facebook recruitment. Additionally, we have discussed the algorithmic bias and the measurement of survey interest more explicitly to further enhance the interpretation of our findings.

---

## [Decision Letter · Decision Letter 2]

15 May 2025

PONE-D-24-35721R2Assessing self-selection biases in Facebook-recruited online surveys: Evidence from the COVID-19 Health Behavior SurveyPLOS ONE

Dear Dr. Donzowa,

Thank you for submitting your manuscript to PLOS ONE. After careful consideration, we feel that it has merit but does not fully meet PLOS ONE’s publication criteria as it currently stands. Therefore, we invite you to submit a revised version of the manuscript that addresses the points raised during the review process.

We look forward to receiving your revised manuscript.

Kind regards,

Anat Gesser-Edelsburg, Ph.D.

Academic Editor

PLOS ONE

**Journal Requirements:**

Reviewers' comments:

Reviewer's Responses to Questions

**Comments to the Author**

1. If the authors have adequately addressed your comments raised in a previous round of review and you feel that this manuscript is now acceptable for publication, you may indicate that here to bypass the “Comments to the Author” section, enter your conflict of interest statement in the “Confidential to Editor” section, and submit your "Accept" recommendation.

Reviewer #3: (No Response)

2. Is the manuscript technically sound, and do the data support the conclusions?

Reviewer #3: Yes

3. Has the statistical analysis been performed appropriately and rigorously? 

Reviewer #3: Yes

4. Have the authors made all data underlying the findings in their manuscript fully available?

Reviewer #3: (No Response)

5. Is the manuscript presented in an intelligible fashion and written in standard English?

Reviewer #3: Yes

6. Review Comments to the Author

**Reviewer #3:** I appreciate the authors’ thoughtful and thorough revisions to the manuscript. The updated version demonstrates meaningful improvements in clarity, methodological transparency, and contextualization of findings. The manuscript now offers a more precise contribution to the literature on online survey recruitment and selection bias, particularly as it pertains to Facebook.

Strengths of the Revision:

The updated title and discussion now appropriately reflect that findings are specific to Facebook-recruited samples.

The authors have added valuable discussion of the limitations introduced by Facebook’s ad delivery system, including a new analysis of ad impressions by demographic group.

The paper now clearly acknowledges the absence of a direct measure of pre-survey interest and offers reasonable suggestions for future studies to address this gap.

The presentation of the modeling strategy, outcomes, and supplementary tables has been improved and is easier to follow.

The authors have responded well to concerns about the generalizability of findings across countries and health contexts, noting the potential influence of media coverage and national policies.

Remaining Concerns and Suggestions:

Generalizability Beyond Facebook:

While the authors now frame the study more narrowly, some claims in the abstract and discussion could still suggest broader relevance to “online surveys” generally. I recommend softening these statements further to reinforce that the results may not translate to other platforms with different audience structures or ad delivery systems.

Unmeasured Motivation and Interest:

The authors acknowledge they did not measure respondents’ interest in COVID-19 or their reaction to specific ad images. This limits interpretability of the observed differences in survey participation across image types. Future studies could address this by including motivation-related questions or pre-randomizing image exposure more rigorously.

Opaque Ad Delivery Mechanism:

Despite additional analyses, Facebook’s ad optimization algorithm remains a non-transparent influence on who sees what ad. While the authors have adjusted for some of this bias through demographic controls and modeling, the underlying ad delivery mechanism still introduces uncertainty into causal interpretation. Running separate ad campaigns per image, as the authors suggest, would be a stronger solution.

Crisis-Specific Context:

Given the timing of the survey during the COVID-19 pandemic, user engagement with health content was likely heightened. This context should be more clearly noted as limiting generalizability to other public health topics or less urgent health issues.

Conclusion:

Overall, this revised manuscript represents a substantial improvement and makes a valuable contribution to the literature on digital recruitment and selection bias. While some methodological limitations remain, they are now better acknowledged and reasonably addressed. I am satisfied with the revisions and would support publication after minor final refinements.

7. PLOS authors have the option to publish the peer review history of their article (what does this mean?). If published, this will include your full peer review and any attached files.

Reviewer #3: No

---

## [Author Response · Author response to Decision Letter 3]

4 Jun 2025

Reviewer(s)' Comments to Author:

Reviewer 3:

I appreciate the authors’ thoughtful and thorough revisions to the manuscript. The updated version demonstrates meaningful improvements in clarity, methodological transparency, and contextualization of findings. The manuscript now offers a more precise contribution to the literature on online survey recruitment and selection bias, particularly as it pertains to Facebook.

We thank the reviewer for the positive feedback and recognition of our efforts in revising the manuscript. We appreciate that the revisions were seen as meaningful improvements in clarity, methodological transparency, and contextualization of our findings. We appreciate the reviewers' time and careful review throughout the revision process.

Strengths of the Revision:

The updated title and discussion now appropriately reflect that findings are specific to Facebook-recruited samples. The authors have added valuable discussion of the limitations introduced by Facebook’s ad delivery system, including a new analysis of ad impressions by demographic group. The paper now clearly acknowledges the absence of a direct measure of pre-survey interest and offers reasonable suggestions for future studies to address this gap. The presentation of the modeling strategy, outcomes, and supplementary tables has been improved and is easier to follow. The authors have responded well to concerns about the generalizability of findings across countries and health contexts, noting the potential influence of media coverage and national policies.

We thank the reviewer for their encouraging comments. We appreciate the recognition of the revised title and discussion, of the added analysis on Facebook’s ad delivery system, and of the clarification of limitations, such as the lack of a direct measure of pre-survey interest. We also appreciate the positive feedback on our improved modeling presentation and responses to generalizability concerns.

Remaining Concerns and Suggestions:

Generalizability Beyond Facebook:

While the authors now frame the study more narrowly, some claims in the abstract and discussion could still suggest broader relevance to “online surveys” generally. I recommend softening these statements further to reinforce that the results may not translate to other platforms with different audience structures or ad delivery systems.

We thank the reviewer for this comment. In response, we have carefully revised the abstract and discussion sections to more clearly reflect the platform-specific nature of our findings. Specifically, we have emphasized that our results pertain to Facebook-based recruitment. The revised version now reads as follow:

- Abstract:

"Overall, our findings do not provide consistent evidence that higher topic salience in our Facebook-based recruitment materials systematically influenced survey responses. However, in specific countries, certain recruitment images were linked to variations in COVID-19 threat perception and uptake of preventive behaviors. These context-specific effects highlight the importance of careful recruitment design for Facebook-based surveys during health crises."

- Lines 587-590:

“Our findings provide limited support for our initial hypothesis that, in the context of Facebook-based recruitment, higher topic salience in the survey recruitment materials would lead to systematically different survey responses.”

- Lines 602-605:

“In conclusion, while topical self-selection had a limited overall impact, context-specific image effects suggest that careful recruitment design remains important for Facebook recruited surveys during health crises.”

Unmeasured Motivation and Interest:

The authors acknowledge they did not measure respondents’ interest in COVID-19 or their reaction to specific ad images. This limits interpretability of the observed differences in survey participation across image types. Future studies could address this by including motivation-related questions or pre-randomizing image exposure more rigorously.

We thank the reviewer for highlighting this important point. We agree that the absence of direct measures of respondent motivation and interest limits the interpretability of differences in participation across image types. In response, we have expanded the suggestions of how future research could address this point. In the revised limitation section it now reads:

Lines 543-545:

“Future studies could address this issue by including a question about respondents' interest in the survey topic or motivation to participate. “

Lines 549-550:

“In addition, the A/B testing feature in the campaign setup could be used to randomize the display of images.”

Opaque Ad Delivery Mechanism:

Despite additional analyses, Facebook’s ad optimization algorithm remains a non-transparent influence on who sees what ad. While the authors have adjusted for some of this bias through demographic controls and modeling, the underlying ad delivery mechanism still introduces uncertainty into causal interpretation. Running separate ad campaigns per image, as the authors suggest, would be a stronger solution.

We thank the reviewer for this thoughtful observation. We agree that Facebook's opaque ad delivery algorithm remains a significant source of uncertainty. In response, we have now highlighted this limitation more clearly in our limitations section before suggesting separate ad campaigns as a potential strategy for future studies.

Lines 569-570:

“However, it is important to acknowledge that the ad optimization algorithm remains opaque to researchers outside of Meta.”

Crisis-Specific Context:

Given the timing of the survey during the COVID-19 pandemic, user engagement with health content was likely heightened. This context should be more clearly noted as limiting generalizability to other public health topics or less urgent health issues.

We thank the reviewer for raising this point. We agree that the heightened public attention to health during the COVID-19 pandemic likely influenced user engagement with our survey content, as we also state at the beginning of our limitations section where we describe the timing of our data collection. In response to the reviewer's comment, we have added a sentence to this section that explicitly states the limited generalizability to other public health topics or less pressing health issues. On lines 527-529 of the revised manuscript, it now reads:

“This may limit the generalizability of our findings to other public health topics or to contexts with less urgent health threats.”

We hope that this addition more clearly communicates the implications of this limitation for interpreting and generalizing our findings.

Conclusion:

Overall, this revised manuscript represents a substantial improvement and makes a valuable contribution to the literature on digital recruitment and selection bias. While some methodological limitations remain, they are now better acknowledged and reasonably addressed. I am satisfied with the revisions and would support publication after minor final refinements.

We thank the reviewer for their thoughtful and constructive feedback throughout the review process. We are grateful for the recognition of the improvements made and are pleased that the manuscript is now seen as a valuable contribution to the literature on digital recruitment and selection bias.

---

## [Editor Report · Decision Letter 3]

6 Jun 2025

Assessing self-selection biases in Facebook-recruited online surveys: Evidence from the COVID-19 Health Behavior Survey

PONE-D-24-35721R3

Dear Dr. Donzowa,

We’re pleased to inform you that your manuscript has been judged scientifically suitable for publication and will be formally accepted for publication once it meets all outstanding technical requirements.

Kind regards,

Prof. Anat Gesser-Edelsburg, Ph.D.

Academic Editor

PLOS ONE
---

## [Editor Report · Acceptance letter]

PONE-D-24-35721R3

PLOS ONE

Dear Dr. Donzowa,

I'm pleased to inform you that your manuscript has been deemed suitable for publication in PLOS ONE. Congratulations! Your manuscript is now being handed over to our production team.

Kind regards,

on behalf of

Prof. Anat Gesser-Edelsburg

Academic Editor

PLOS ONE